# Analysis of the Diversity Presented by *Vitis vinifera* L. in the Volcanic Island of La Gomera (Canary Archipelago, Spain) Using Simple Sequence Repeats (SSRs) as Molecular Markers

Francesca Fort [1,*,†], Qiying Lin-Yang [1,†], Carla Valls [1], Pau Sancho-Galán [2], Joan Miquel Canals [1] and Fernando Zamora [1]

1   Grupo de Tecnología Enológica (TECNENOL), Department of Biochemistry and Biotechnology, Faculty of Oenology, Rovira i Virgili University, Sescelades Campus, C/Marcel lí Domingo, 1, E-43007-Tarragona, Spain; qiying.lin@estudiants.urv.cat (Q.L.-Y.); carlavlls813@gmail.com (C.V.); jmcanals@urv.cat (J.M.C.); fernando.zamora@urv.cat (F.Z.)
2   Department of Chemical Engineering and Food Technology, Science Faculty, University of Cadiz, Agrifood Campus of International Excellence (CeiA3), Wine and Food Research Institute (IVAGRO), P.O. Box 40, 11510 Puerto Real, Spain; pau.sancho@uca.es
*   Correspondence: mariafrancesca.fort@urv.cat
†   These authors contributed equally to this work.

**Abstract:** La Gomera Island is one of the areas of our planet where the phylloxera plague never arrived. To measure the genetic diversity of the vine after more than 500 years (inter- and intravarietal variability) of adaptation to this new environment, a prospection was carried out. For this purpose, 120 samples were collected and genotyped with 20 SSRs. A total of 52 unique profiles were found corresponding to 4 new varieties (Coello blanca, Barrerita negra, Malvasia periquin gomerae, Verdello gomerae), 9 individuals identical to the most widespread profile, and 39 individuals that presented variations (1 corresponding to a mutation of a new variety (Verdello gomerae de Monacal) and 38 corresponding to variations of known varieties, some of which included cases of triallelism or quadriallelism). The population of local vines in La Gomera Island is considered to be the most unique in the Canary Islands to date. It is hypothesised that the grapevine varieties Malvasia periquin gomerae and Verdello gomerae are possibly the most unique and that the Barrerita negra variety may have resulted from an interspecific crossbreeding. The Coello blanco variety (admixed) seems to have a strong Central European influence. Finally, we propose that the prime name for the Albillo forastero variety, which was arbitrarily imposed by the scientific community, be changed to the more widespread and better-known name in La Gomera Island and the Canary Archipelago, which is Forastera gomerae.

**Keywords:** *Vitis vinifera* L.; SSR; microsatellite; diversity; volcanic; Canary Islands; La Gomera

## 1. Introduction

Nowadays, the cultivation of *Vitis vinifera* L. varieties is one of the oldest and most relevant worldwide. The FAO (Food and Agriculture Organization of the United Nations, Washington, DC, USA) affirms that the most economically valued crop is the vine for the vinification of medium- and high-quality wines [1]. Furthermore, the NASS (National Agricultural Statistics Service) reaffirmed its importance as the sixth most economically valuable crop in 2021 in the United States [2]. The global production volume of fresh grapes in 2021, according to the OIV (International Organisation of Vine and Wine), was 74.8 million tons, of which more than 20% came from Europe. Specifically, and in this order, Italy, Spain, and France led European production by a wide margin, representing 79% of the total production [3].

However, if one examines the cultivated varieties' biodiversity on a global scale, it can be seen that almost the same varieties are grown in most countries, occupying between

70 and 90% of the cultivated area. In this sense, these varieties are already known as international varieties [1]. This fact highlights the need for further exploration and study of the rest of the varieties. It is, therefore, about searching for unique and interesting molecular profiles (MP-SSRs) that not only help to expand the range of wines offered in the current market or mitigate the effects of climate change but also give advantages to other relevant factors, whether biotic (pests and/or diseases) and/or abiotic (salinity, etc.).

A starting point for this purpose is undoubtedly La Gomera Island. This island is located in European overseas territory (near Western Sahara); it belongs to the volcanic archipelago of the Canary Islands and Macaronesia (a group of Atlantic volcanic islands) (Figure 1) [4].

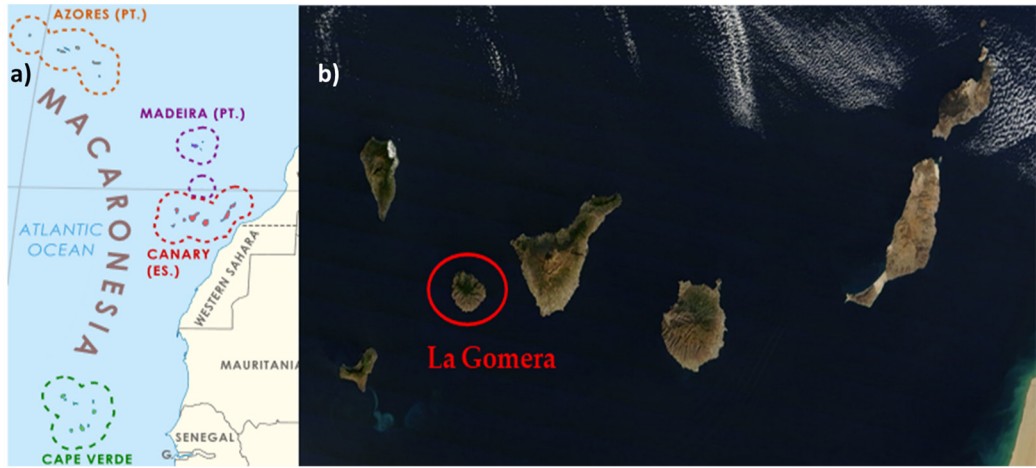

**Figure 1.** (**a**) Macaronesia map [5]. (**b**) Canary Islands archipelago and La Gomera Island [6].

It is now widely accepted that the vine was not a part of the existing crops in the Canary Islands until its introduction in the 15th century. This belief was challenged by the discovery of fossilised *Vitis* seeds in archaeological excavations [7]. However, *Vitis vinifera* ssp. *sylvestris*, like much of the endemic and fossilised flora of the islands, does not appear today, as it disappeared afterward for unknown reasons. It is for this reason that grapevines in the Canary Islands come from those cultivated (domesticated) varieties introduced by Europeans. Thus, a low diversity of vines compared to other regions of the planet could be expected, but the opposite is true. Due to the phylloxera plague (*Daktulosphaira vitifoliae*) that hit Europe at the end of the 19th century, the loss of *Vitis vinifera* L. varieties was much more accentuated on the mainland than in the Canary Islands, which remained unaffected by this plague. It is for this reason that the Canary Islands, including La Gomera Island, are one of the last strongholds of the European vine varieties that existed before this plague [8]. As a result, vines have evolved and adapted to this new habitat for more than 500 years (through natural crosses, mutations, and selection (natural and anthropogenic)).

La Gomera Island has a similar climate to the islands of La Palma and El Hierro (hot, semiarid climates, BSh Köppen) with incident precipitation in high areas and scarce precipitation near the coast [9]. Furthermore, if its orography is observed, it can be concluded that apart from making possible the humid microclimate of Garajonay Park (laurel forest dating from the Tertiary period, which is also a World Heritage Site), the only areas suitable for agricultural cultivation are the mountain's slopes and ravines [8]. These areas were initially unsuitable for vine cultivation. In these circumstances, the traditional farmers overcame this problem with heroic viticulture based on the construction of dry-stone terraces (Figure 2). These were staggered stone constructions without cement (mortar) on the slopes, which made planting possible. Under these conditions, when the crops took root, they strengthened the construction and prevented the soil from sliding away [10].

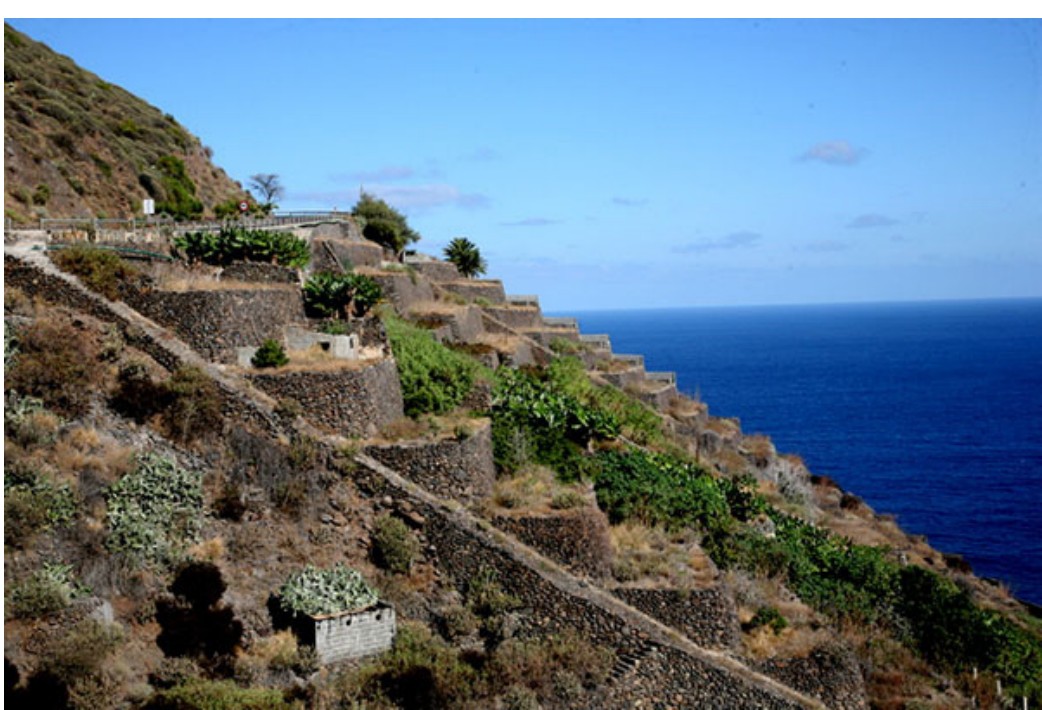

**Figure 2.** Staggered dry-stone terrace in La Gomera [10].

Grape-berry production in La Gomera Island is currently one of the most important crops in the island's agricultural sector. Within the area for permanent crops (citrus and noncitrus fruit trees, vineyards, olives, etc.), 139.7 hectares are reserved for dry crops. Specifically, 117.9 hectares are reserved for vine cultivation, almost all of which are used for winemaking [11]. In 2020, the total grape production was 53,576 kg, of which 38,834 kg corresponded to white grapes. Examined by variety, the Albillo forastero is the most important variety in this category. Comprising more than 82% of the total amount of white grapes of the island [12] and 85% of the wine-growing area of the island of La Gomera, Albillo forastero is the most cultivated grapevine variety [13]. It is currently known that this variety comes from a cross between the Andalusian variety Palomino fino (known in the Canary Islands as Listan blanco) and the Portuguese variety Verdelho branco (known in the Canary Islands as Verdello). This cross also produced another Canary Islands variety known as Albillo criollo (a local variety from La Palma Island) [14].

The island of La Gomera has had its own Appellation d'Origine Contrôlée (AOC) since 2003, which accepts 31 varieties for winemaking [12]. The most curious thing is that among the accepted varieties, there is no Albillo forastero, the prime name (PN) found in the *Vitis* International Variety Catalogue (VIVC) [15], a worldwide database for grapevine varieties. On the other hand, the synonymous names of this variety, Forastera blanca and Forastera gomera, are found. Moreover, on La Gomera Island and in the Canary Archipelago (the only place in the world where this variety can be found), the name Albillo forastero has never been used to refer to this local variety. It is also observed that the term "local" is always used and not the term "autochthonous". Accepting the proposal of Dr. Manna Crespan (2nd OENOVITI International Symposium, 2014) [16], the term autochthonous is not used because the main variety of the island of La Gomera did not originate on the island itself, but it is the island's inhabitants who have preserved it until the present day, making it unique in the world.

Historically, grapevine varieties' characterisation and identification were based on the description and comparison of morphological characters. The science that allowed these comparative studies of *Vitis vinifera* L.'s phenotypes to be carried out is ampelography [17]. Two of the most widespread errors generated by this phenotypic science (which is markedly subjective, influenced by the abiotic factors of the environment and by the state of health

of the individual to be analysed) are homonyms and synonyms. A term is said to be a homonym when it is used to name two or more vine varieties. This is the case for the term Ugni blanc, which is used to name three different varieties (the Italian variety Trebiano toscano, the Spanish variety Viura, and the Portuguese variety Douradinha). On the other hand, a term is qualified as a synonym when it is one of the different names of the same variety. In this case, a variety has a PN and one or more synonymous names. As an example to illustrate this concept, for the most cultivated variety on La Gomera Island, the PN is Albillo forastero and the four synonymous names registered in the VIVC are Forastera, Forastera blanca, Forastera gomera, and Gomera [15]. Of all the names by which the same variety is known, the most widespread or the one most commonly used in the place where the variety possibly originated or where it has been preserved is chosen as the PN [18,19]. This is not the case in this example. The names most commonly used on La Gomera Island and in the Canary Islands are Forastera blanca and Forastera gomera, and it is the international community that has arbitrarily imposed the term Albillo forastero, unknown to all the inhabitants of La Gomera and the rest of the Canary Islands, as the PN. In order to avoid the appearance of errors of this and other kinds, from the 1990s onwards, different molecular marker techniques were implemented for the characterisation and identification of grapevine varieties based on the study of the genotype (which is practically invariable). Currently, the most widely implemented and therefore used are SSRs (simple sequence repeats) or microsatellites and SNPs (single-nucleotide polymorphisms) [19].

This work aimed to study the intervarietal and intravarietal diversity of the island of La Gomera existing at the time of sampling using the SSR technique. In addition, another objective was to detect possible errors in terminology and to find synonyms and homonyms, if any. The last objective was to verify the uniqueness of the population of the La Gomera grapevine population. For this purpose, a population structure study was carried out, comparing the population of local varieties of La Gomera with those of other populations in the Canary Islands and the rest of the world.

## 2. Materials and Methods

### 2.1. Plant Material

In order to explore the *Vitis vinifera* L. biodiversity extent, 120 vine shoot samples were collected in different municipalities of La Gomera Island (Agulo, Alajeró, Hermigua, San Sebastián de La Gomera, Valle Gran Rey, and Vallehermoso) through a mass selection process carried out by the local winegrowers and supervised by the Regulatory Council of the Vinos de La Gomera AOC technicians. Once harvested, shoots were stored at −20 °C until processing. Table S1 shows detailed information on the analysed grape varieties.

### 2.2. DNA Extraction and Purification

The purified DNA was obtained using a specific method for foliar and recalcitrant tissues such as wood and seeds [20,21]. This is an adaptation based on the protocol of Fort et al. [22], which was improved by adding PVP (polyvinylpyrrolidone) to the extraction buffer and by adding two chloroform washes. With the help of a Thermo Fisher® Scientific NanoDrop TM 1000 spectrophotometer and using the electrophoresis technique, the levels of purity, integrity, and concentration of the nucleic acid were precisely evaluated.

### 2.3. Microsatellites

The twenty SSRs that the Investigación en Tecnología Enológica (TECNENOL) research group uses for genotyping samples from the Canary Islands were also tested on this occasion (VVS2, VVS3, VVS29 [23]; VVMD5, VVMD6, VVMD7 [24]; VVMD27, VVMD28, VVMD36 [25]; VrZAG21, VrZAG47, VrZAG62, VrZAG64, VrZAG79, VrZAG83 [26]; VvUCH11, VvUCH12, VvUCH19 [27]; scu06vv [28]; VChr19a [29]). Of these, there are two SSRs that are not independent, as they analyse the same area of the genome with different primers. These are VrZAG47 and VVMD27 [30]. Furthermore, in this SSR "kit" used by TECNENOL, there are 7 SSRs that coincide with some of the 9 SSRs accepted by the international scientific

community [31]. In Table S2, the values of the allelic lengths of the SSRs considered as international standards corresponding to the unique MP-SSRs of this grapevine population can be seen.

### 2.4. DNA Amplification

For the satellite region amplification, an AmpliTaq DNA Polymerase kit (Applied Biosystems, Foster City, California, USA) was used with a final reaction volume of 12.5 μL. The amounts of each reagent broken down per well were as follows: 1.25 μL of buffer, 2 μL of dNTPs, 0.125 μL of deionised formamide, 0.0625 μL of Taq polymerase, and 4 ng of DNA. In addition to 1 μM of each primer (1.25 μL), with the particularity that the forward primer (Fw) was labelled with a specific fluorochrome (6-FAM: VVS3, VVMD7, VVMD28, VVMD36, VrZAG47, VrZAG62, VrZAG83, VvUCH11, and VvUCH19; HEX: VVS2, VVS29, VVMD6, VVMD27, VrZAG21, VrZAG79, and VChr19a; NED: VVMD5, VrZAG64, scu06vv, VvUCH12). Seven thermocycling blocks were performed according to the different annealing temperatures (*Ta*) (Table S3), and for all of them, the thermocycling conditions were as follows: (a) a first stage of 5 min at 95 °C; (b) a second stage of 40 cycles: 45 s at 95 °C during, 30 s at the corresponding Ta, and 1 min 30 s at 72 °C; and (c) a third stage of 7 min at 72 °C. An Applied Biosystems 2720 Thermal Cycler was used for this process (Foster City, CA, USA).

### 2.5. Amplified Fragments Length Measurement

Measurements of the polymerase chain reaction (PCR) amplified fragments was performed by capillary electrophoresis with an ABI PRISM 3730® genetic analyser (Applied Biosystems, Foster City, CA, USA). The preparation of the amplified plates was carried out by adding to each well the corresponding amplification product, 20.5 μL of deionised formamide, and 0.25 μL of the internal marker GeneScan ROXTM 500 (Applied Biosystems, Foster City, CA, USA). Each plate was then denatured by a thermocycling regime at 95 °C for 3 min. Peak Scanner Software 2.0 (Applied Biosystems, NJ, USA) was used to size the amplified fragments.

### 2.6. Data Analysis

GenAlEx 6.5 software was used for different purposes [32,33]. Data input files were prepared according to this program using the Excel program 2016 (Microsoft 365, USA). Firstly, the efficiency and effectiveness of the SSR kit used by TECNENOL was evaluated. For this purpose, the following statistical parameters were measured: Na (number of different alleles), Ne (number of effective alleles: alleles that are transmitted to the next generation), Ho (observed heterozygosity: the computed heterozygotes), diversity index or He (expected heterozygosity: estimation of the heterozygotes that the population under study could have), F (fixation index: parameter that measures the goodness of homozygotes), and PI (probability of identity: the probability that two MP-SSRs with the same SSR value are the same variant). Secondly, it allowed us to find all the MP-SSRs that matched each other in order to eliminate redundant information. Assignment tests to check the sample distribution goodness of fit in different populations were also performed using this program. GenAlEx 6.5 bases this strategy on the allele frequency of each accession. It also allowed us to calculate a logarithmic probability value of this accession for each subpopulation using the allele frequencies of the respective subpopulations and assign an individual to the subpopulation with the highest logarithmic probability value [34]. Thirdly, based on the standardised covariance of the genetic distances calculated for the codominant markers, GenAlEx 6.5 enabled two-dimensional graphical representations to be made for a set of populations and also for a set of individuals belonging to different populations. Finally, it made it possible to calculate the coefficient of genetic differentiation between populations assuming the infinite allele model (Fst).

Python Data [35], applying Matplotlib strategy, and MEGA version 7 [36], applying the neighbour-joining approach [37], were used to make the three-dimensional graphical representations of PCoA, the phylogenetic trees, and the circular dendrograms.

The structure of the different populations that emerged from this study was explored using the cluster analysis method based on models implemented in the program Structure 2.3 [38,39]. This program calculates a probability value for a predetermined number of K populations (or clusters) and assigns the part of each individual's genome derived from each cluster. Population structure was tested from K = 1 to K = 7 for the local population of La Gomera Island and for the Canary Islands population and from K = 1 to K = 9 for the world population. Ten independent replications were performed, composed of 1.000.000 Markov chain Monte Carlo (MCMC) steps after discarding the first 100.000. It was assumed that the current populations' allele frequencies were correlated and that they could have originated from more than one ancestral population. The most probable value of K was determined according to the method of Evanno et al. [40]. The parameter *q* defines what proportion of an individual's genome belongs to the different predefined clusters (K). The membership of a population to a cluster was accepted for mean values of $q \geq 0.85$.

## 3. Results

### 3.1. SSR Polymorphism

Once the 120 samples were genotyped, the first data normalisation was carried out by eliminating 68 identical profiles (including the 2 "sports": Bermejuela negra and Malvasia rosada) (Tables S1 and S2). From the remaining 52 MP-SSRs, which corresponded to 19 varieties, the relevant statistics were calculated to determine the goodness of fit of the 20 SSRs used. Table S4 presents the results of the main statistics. The total number of alleles in the unique MP-SSR population was 181, with a mean value of 9.1. The SSRs with the highest number of alleles were VVMD28 with 16 alleles and VVMD36 with 13 alleles. The SSRs with the lowest number of alleles were VVS3 with four alleles, UCH19 with five alleles, and VVS29 with six alleles. The mean number of effective alleles was 4.4, with the SSRs VVMD28, VVMD27, and VVMD36 showing the best values (8.32, 6.76, and 6.72, respectively), and the worst results were shown by the SSRs VVS29 with 1.41 alleles, VVS3 with 2.17 alleles, and VVS3 with 2.17 alleles passing to the next generation, respectively. The mean Ho (78.8% heterozygotes) was higher than the He (72.4%). The highest percentages of heterozygotes were observed in the SSRs VVMD36 with 98.1% heterozygotes; VVS2 with 96.2%; and ZAG79, VVMD28, and SCU06 with 94.2%. In contrast, the lowest values corresponded to VVS29 with 30.8%, UCH19 with 51.9%, and VVS3 with 55.8%. The highest values of the diversity index, also known as He, were for SSRs VVMD28, ZAG47, VVMD27, and VVMD36 (88%, 86.9%, 85.2%, and 85.1%, respectively), and the lowest values were again for SSRs VVS29 with 29.1%, UCH19 with 51.9%, and VVS3 with 53.9%. Only four SSRs had an F index with zero or positive values; these were UCH19 (F = 0.000), ZAG47 (F = 0.005), ZAG83 (F = 0.112), and UCH12 (F = 0.144). The lowest PI was shown by SSRs VVMD28, ZAG47, VVMD27, and VVMD36 with values of $2.55 \times 10^{-2}$, $3.04 \times 10^{-2}$, $3.86 \times 10^{-2}$, and $3.95 \times 10^{-2}$, respectively, and the highest PI was shown by SSRs VVS29, VVS3, and UCH19 with values of $5.16 \times 10^{-1}$, $3.07 \times 10^{-1}$, and $2.68 \times 10^{-1}$. The accumulative PI for the 20-SSR kit reached a value of $3.1 \times 10^{-21}$.

### 3.2. Grapevine Variety Analysis

All of the original and conclusive information concerning the 120 accessions can be found in Table S1. In addition, it presents in detail the similarity of the MP-SSR of a given accession concerning the most widespread MP-SSR according to the TECNENOL database, even specifying which allele presents the variation. With all of this information and the possibility of comparing it with the VIVC database, a study was carried out at both the molecular and terminological scales. In Table S2 and for the 52 unique MP-SSRs, the numerical values of the allelic lengths measured for the 7 international SSRs available to the TECNENOL research group are also presented. Furthermore, the accessions in

these tables (Tables S1 and S2) correspond to 19 varieties, of which 5 are local varieties from the island of La Gomera (Albillo forastero, Barrerita negra, Coello blanca, Malvasia periquin gomerae, Verdello gomerae); also, 5 are local varieties from the rest of the Canary Archipelago (Bermejuela, Listan negro, Torrontes volcanico, Uva de Año, Verijadiego), and the remaining 9 correspond to foreign varieties from the Canary Islands: 3 from Spain (Mollar cano, Palomino Fino, Tempranillo tinto), 2 from Portugal (Caracol, Verdelho branco), 1 from France (Alicante Henri Bouschet), 1 from Greece (Muscat of Alexandria), 1 from the Balkan Peninsula (Malvasia Dubrovacka), and 1 from the United States of America (Ruby cabernet). Mutations should also be highlighted, whether colour (sport) or numerical (see Table S1). There is a colour mutation that corresponds to a sport widely spread throughout the Canary Islands (Malvasia Dubrovacka rosada), another that corresponds to a sport from the island of La Gomera (Bermejuela negra), and an individual that presents a numerical variation with respect to a variety from the island of La Gomera (Verdello gomerae de Monacal).

In both Tables S1 and S2, it can be seen that the grouping corresponding to the Albillo forastero variety is made up of 46 individuals and is, therefore, the most numerous. Among them, 26 accessions were identical to the most widespread MP-SSRs in the TECNENOL database (identities), 16 individuals presented variations in their allelic length in one allele, 3 accessions showed variations in two alleles, and 1 sample presented variation in three alleles. These results defined 10 different MP-SSRs for this variety. One MP-SSR corresponded to an identity with the most widespread profile in our database. This profile was named with its PN according to the VIVC, which in this case corresponded to the term Albillo forastero. Six MP-SSRs showed allelic length variation in one allele: (a) The variation in VVS3-1 (numerical change in the first SSR allele VVS3), which is known as Forastera de la Isla Redonda and was previously published by Fort et al. [41]. (b) The VVS3-2 variation that was called Forastera blanca de Agulo. (c) The mutation in VVS29-2, which is known as Forastera blanca de Vallehermoso. In addition, there are two mutated profiles in this same allele but with different allelic lengths that are marked with *, but all are known with the same term. (d) The variation in UCH12-2 known in La Gomera as Forastera blanca roquillos. Two individuals showed variation in their allelic length in two alleles of the same SSR (VVS3-1, VVS29-2), but as in the previous case, a numerical difference in allelic length was detected for VVS29-2. Both specimens are known as Forastera blanca Simancas. The last profile detected for the variety Albillo forastero corresponded to an accession with variation in allelic length in three alleles (VVS3-1, VVS29-1, VVS29-2) known on the island of La Gomera as Forastera blanca tamargada.

The only representative of the French variety Alicante Henri Bouschet, known in La Gomera as Alicante tintilla, had a mutation in one allele (VVS29-2).

The local Canarian variety Bermejuela cluster consisted of seven members, two identities, two mutations in one allele, one mutation in two alleles, one mutation in four alleles, and a colour mutation with the same MP-SSR as the most widespread genetic profile in the TECNENOL database (these cases are known as "sport"). This colour mutation is known in La Gomera Island as Marmajuelo negro [14]. This grouping provided four different MP-SSRs: (a) the identity itself known by the name Bermejuela; (b) the variation in one allele (VVS3-2), a profile already described by TECNENOL in El Hierro Island and named Bermajuelo del Echedo [41], which is also known on the island of La Gomera by the synonymous name Marmajuelo corto (Table S1); (c) the variation in two alleles (VVS3-2, SCU06-1) known as Marmajuelo de Vallehermoso; and (d) the variation in four alleles (VVS3-1, VVS3-2, VChr19a-1, VChr19a-2) known in La Gomera as Marmajuelo de Valle bajo.

The Portuguese variety Caracol was recorded under the erratic name Forastera negra and did not show any variation in its MP-SSR.

There are eighteen entries of the local Canarian variety Listan negro, of which half are identities, eight have one variation in one allele, and the last one is mutated in two alleles. This group of individuals gives rise to six different MP-SSRs, from the identity

named Listan negro to the profiles mutated on one allele which give rise to four different MP-SSRs: (a) variation in VVS3-1 which is known as Listan negro de Hermigua; (b) two variations, one in VVS2-2, for which the profile is already published under the name Listan negro santanero [42], and one in VVS29-2, for which the profile is also published under the name Listan negro de la corona [42]; (c) variation in VVMD6-1 known in La Gomera as Listan negro de Vallehermoso; and finally (d) the accession registered as Listan negro de lo Machado, which corresponds to an MP-SSR with variations in VVS3-1 and ZAG83, the last variation being a case of triallelism.

The five members of the Malvasia Dubrovacka variety are distributed in one identity, two individuals mutated in one allele, and another mutated in three alleles of which one is triallelic; the last component, a rosé sport very widespread in the Canary Islands and unique in the world, is the Malvasia rosada. For this variety, there are four MP-SSRs corresponding to an identity known as Malvasia Dubrovacka, a variation in VVS2-2 known as Malvasia blanca de Agulo, a variation in VVS29-2 known as Malvasia blanca de Vallehermoso, and finally the Malvasia blanca piedra gorda with three variations in VVS3-2, VVS29-2, and SCU06, the latter variation being a case of triallelism.

In La Gomera Island, three specimens of the Mollar cano variety were collected, one identity with the same name and two individuals with one variation in one allele. These were the Negramoll de Vallehermoso (VVS3-1) and the Mulata del macayo (SCU06-2) mutations.

In contrast, for the Muscat of Alexandria cluster, only individuals with variations were present. Two accessions with one variation in one allele and two with variations in two alleles displayed a total of three MP-SSRs, with one variation being the accession Moscatel de Hermigua (VVS29-2) and two variations being the mutations Moscatel de la caleta (VVS3-1, VVS29-2) and Moscatel de la caleta fino (VVS29-2, SCU06-2).

The grouping of the Palomino fino grapevine variety, known in the Canary Islands by the synonymous name Listan blanco, contained 21 individuals. Of these, 13 were identities. Six had one variation in an allele, and of these, two were triallelic. In addition, one individual had two variations, one of which was triallelic, and the last one had four variations, one of which was also triallelic. In total, six different MP-SSRs were computed: (a) The identity. (b) An accession, Listan blanca chicharrera, with a variation in VVS3-1, which was already published [42]. In addition, Listan blanco de Hermigua, with a variation in VVS3-2 and a variation in the form of triallelism in SSR ZAG83, was already published in the Lanzarote Island prospection as Listan blanco de la bodega [42]. (c) The profile known as Listan blanco de Vallehermoso, presenting variations in VVS29-2 and ZAG83 (triallelic). (d) The entry registered as Listan blanco de espina with four variations (VVS3-1, ZAG79-1, ZAG79-2, ZAG83 (triallelic)).

A single entry corresponded to a Ruby cabernet mutated in three alleles (VVS29-2, VVMD27-1, VVMD36-1) and known as Ruby cabernet ingenio.

The Spanish variety Tempranillo tinto was represented by two mutated individuals, one on an allele (VVS29-2), called Negra mora, and the other, known as Tempranillo de Vallehermoso, with a very rare case of quadriallelism in the SSR (SCU06).

The local variety of Lanzarote Island, Torrontes volcanico, was represented by three accessions, one of which was an identity while the other two had variation in one allele. The mutation known as Torrontes volcanico montoro had a variation in UCH12-2, and Torrontes volcanico machado had a variation in SCU06-1.

The variety also originating from Lanzarote, Uva de año, presented only one component (Uva de año montoro), which presented two variations in two different alleles (VVS3-1, VVS29-2).

With only one component, there were also the following: (a) The Portuguese Verdelho branco with a case of triallelism in the SSR UCH19 (Verdello blanco del Corte). (b) The Verijadiego grapevine variety from El Hierro Island with an identity. (c) Three of the four new varieties: the variety Barrerita negra, also described by Rodríguez-Torres [43] as unknown no. 2; the variety Coello blanca; and Malvasia periquin gomerae, which presented a case of quadriallelism in the SSR (SCU06). (d) The last new variety, which presented two

individuals, of which one was a mutation of the other in two alleles. These were the variety Verdello gomerae and its mutation Verdello gomerae de Monacal (VVS3-1, ZAG83-1).

*3.3. Genetic Structure of the Grapevine Population of the Island of La Gomera*

In order to carry out the study of the grapevine population uniqueness in La Gomera Island, a second data normalisation was carried out. Table S2 shows the 52 unique MP-SSRs corresponding to 19 varieties, which means that 33 individuals were left out because they were genetically very close to each other and could alter the final result of the corresponding study (63.5%). In this sense, the representatives of the 19 varieties mentioned above remained, which did not always correspond to identities, but for the cases of the Alicante Herni Bouschet (with a similarity to the most widespread MP-SSR of 97.5%, i.e., with a variation of only one allele), Muscat of Alexandria (97.5%), Ruby cabernet (92.5% (with a variation of three alleles)), Tempranillo tinto (97.5%), Uva de año (95% (with a variation of two alleles)), and Verdelho branco (97.5%), these were represented by individuals with variations. In order to obtain the best population distribution in different ancestral populations (K), distributions by population from one to seven were tested, with the best distribution being in two populations (K = 2) (Figure S1). Each individual from the population of La Gomera Island was distributed according to a statistical parameter *q* (which indicates the percentage of its inferred genome that belongs to one of these populations) in these two proposed populations. In this study, the arbitrary percentage of 85% [44] was chosen, so that values for $q \geq 85\%$ correspond to pure individuals belonging to a given population and those with a value of $q < 85\%$ are admixed individuals for the same population. Figure 3 shows the best distribution for the La Gomera Island population in two populations (POP1 and POP2). Figure S2 shows the values for the parameter *q* as well as the origin of the members of each population considered. Thus, POP1 consisted of four individuals and represented 21% of the total population on the island. Of these, three individuals were pure, two were new varieties (Malvasia periquin gomerae and Verdello gomerae), and one was the Balkan variety Malvasia Dubrovacka. The population's fourth member, and admixed, was the Spanish variety Tempranillo tinto (97.5%). In contrast, for POP2 (79% of the island population), all individuals were found to be pure. This grouping included (a) three varieties from La Gomera, of which two were new (Barrerita negra and Coello blanca) and one was the well-known Albillo forastero; (b) five Canary Islands varieties (Bermejuela, Listan negro, Torrontes volcanico, Uva de año (95%), and Verijadiego); (c) two Spanish varieties (Mollar cano and Palomino fino); (d) two Portuguese varieties (Caracol and Verdelho branco (97.5%)); (e) one Greek variety (Muscat of Alexandria (97.5%)); (f) one French variety (Alicante Henri Bouschet (97.5%)); and (g) one American variety (Ruby cabernet (92.5%)).

Apart from locating correctly in each population (POP1 and POP2) the different members of the La Gomera Island population, the Structure 2.3 program allowed us to detect the admixed individuals that, in another standardisation, were eliminated, with the aim of constructing the main coordinate plots (PCoA) without interference. Thus, the only admixed individual in both populations, Tempranillo tinto (97.5%), was eliminated, leaving La Gomera Island with 18 components distributed in two populations. An assignment test was carried out using the GenAlEx 6.5 program, resulting in a 100% goodness of fit.

The two- and three-dimensional PCoA representations are presented in Figure 4.

Coordinate 1 (with a goodness of fit of 15.11%) and coordinate 2 (with a goodness of fit of 12.71%) in Figure 4a give a total reliability of 27.82%, separating the 18 grapevine varieties into four quadrants. The upper-right quadrant contains all of the POP1 varieties, with a clear influence from the Eastern Mediterranean Basin (specifically from the Balkan Peninsula (BP)). Of the three POP1 varieties, it is the local ones from the island of La Gomera that are significantly distant from the Malvasia Dubrovacka, located in the lower part of the quadrant. POP2, on the other hand, is distributed throughout the rest of the quadrants. In the upper-left quadrant, the Coello blanca and Verdelho branco varieties (97.5%) are also positioned far away from the rest. In the lower-right quadrant are the Canary Island

varieties Torrontes volcanico and Uva de año (95%), the Spanish Mollar cano, and the American Ruby cabernet. The rest of the POP2 representatives are located in the lower-left quadrant. In the three-dimensional representation of the population of La Gomera, a higher goodness is obtained (36.96%), but the same distribution is maintained with some significant changes (Figure 4b). Two important shifts were detected: the approximation of Muscat of Alexandria closer to the BP-influenced varieties and the approach of the local variety from La Gomera, Albillo forastero, closer to the pairing of Coello blanca and Verdelho branco (97.5%). It should also be noted that in both presentations, the local variety Barrerita negra continued to stay away from the rest of the varieties in its quadrant.

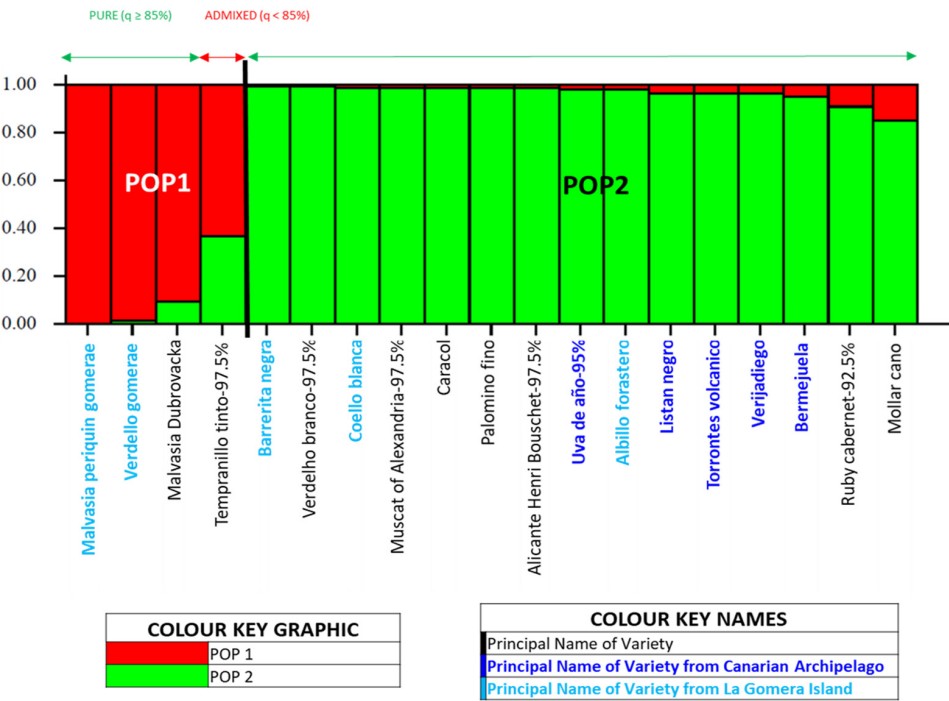

**Figure 3.** La Gomera grapevine varieties' populations (unique MP-SSRs). Structure 3.2 diagram: K = 2 distribution for pure and admixed individuals.

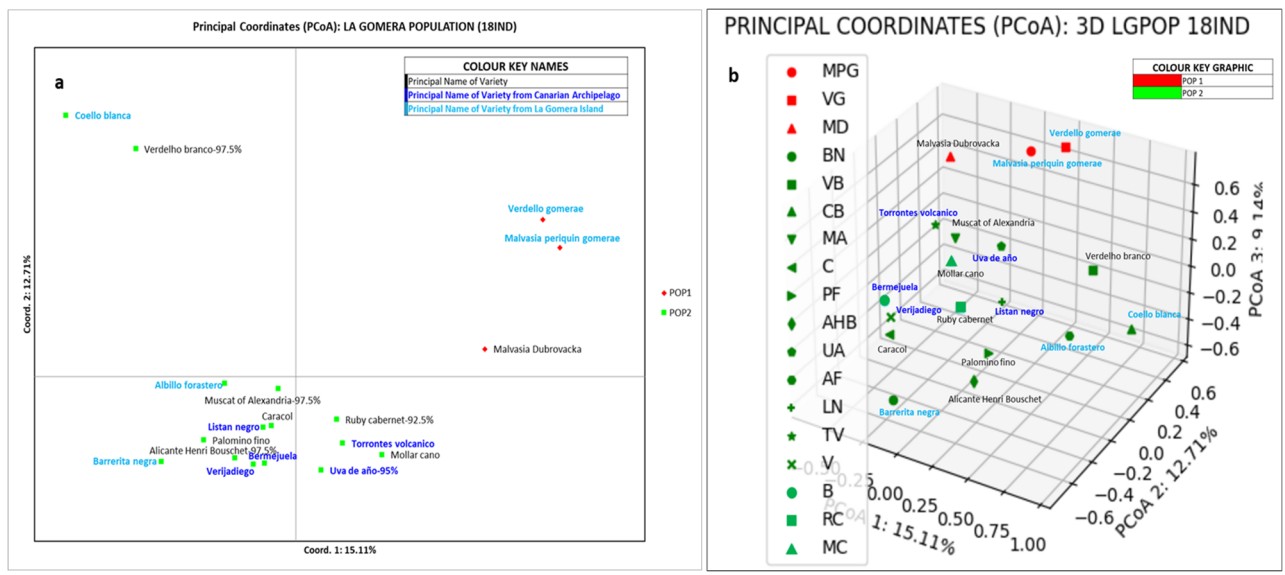

**Figure 4.** PCoA representations of the grapevine varieties' populations from La Gomera Island normalised for K = 2. (**a**) Two-dimensional representation of the 2 populations by individual, and (**b**) three-dimensional representation of the 2 populations by individual.

### 3.4. Relation of La Gomera Grapevine Population with respect to the Canary Archipelago Population

Once the above results were obtained, it was considered appropriate to begin a comparative study of the local varieties of La Gomera Island with respect to the local varieties of the Canary Islands in order to gauge the uniqueness of the population of this first island and, furthermore, to see if any other noteworthy characteristics emerged that should be taken into account. The starting point was a total population of 36 individuals, 5 of which were local varieties from La Gomera (the Albillo forastero variety was added to the 4 new varieties).

The same strategy was used as in the previous section. Thus, the Structure 2.3 program was used to obtain the best distribution for the population of Canary Islands varieties. After testing up to seven different distributions, it was the one corresponding to K = 5, i.e., distributing the population of 36 individuals in five ancestral populations, which gave the smallest error and therefore was the best of all the distributions (Figure S3). Figure 5 shows the graphical representation of the Structure 2.3 program, displaying the distribution of all individuals by ancestral population (as a function of the $q$ statistic) and by pure and admixed varieties. The $q$ numerical values and the statistical data associated with the whole study are shown in Figure S4. Figure 5 and Figure S4 show that POP1 consists of a total of seven varieties (19%) almost equally distributed between the islands of Lanzarote and El Hierro, of which five are pure (71%) and two are admixed (29%). POP2 is made up of nine components that represent 25% of the total Canary Islands population, of which five are pure varieties (56%) and four are admixed (44%). This cluster includes the interspecific cross known as Bienmesabe tinto (Canary Islands (IC)), in addition to the variety from Fuerteventura Island, which has been postulated so far as another possible interspecific cross [45], alongside other varieties from all over the archipelago that usually appear as very singular MP-SSRs in other TECNENOL research group studies. The local Gomeran variety, Barrerita negra (pure), is also included in this grouping. POP3 is constituted by eight members (22%) mainly from Lanzarote Island, which are related very closely or completely to the original variety of BP Malvasia Dubrovacka. It consists of four pure and four admixed individuals. In this group are present as pure individuals the new local varieties of the island of La Gomera, Malvasia periquin gomerae and Verdello gomerae. The POP4 group with six components represents 17% of the total and is formed almost exclusively by varieties from El Hierro Island, of which three are pure and three are admixed. Finally, POP5, also with six components distributed in three pure and three admixed varieties, is characterised by the fact that all the pure individuals come from IC and the two local varieties present are admixed (Albillo forastero and Coello blanca).

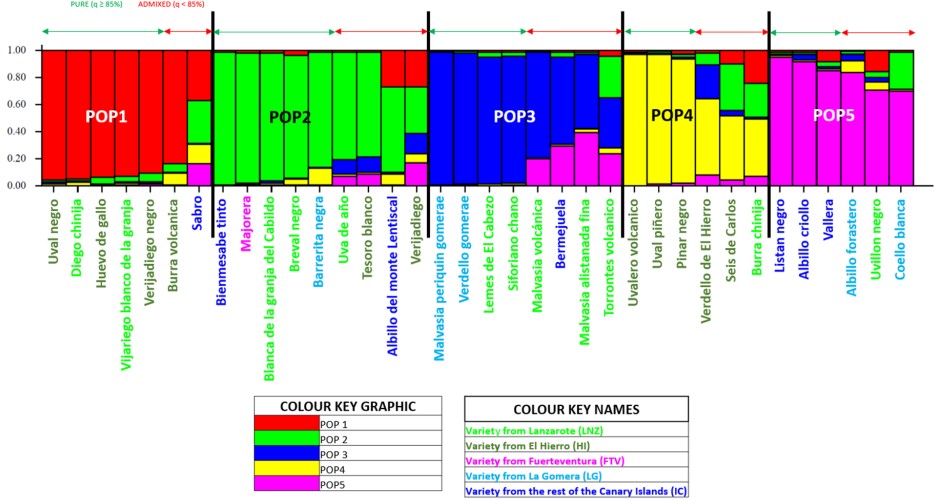

**Figure 5.** Canary grapevine varieties' populations (unique molecular profiles). Structure 2.3 diagram: K = 5 distribution for pure and admixed individuals.

Once the pure varieties were known, the data were standardised by eliminating the admixed varieties, with the aim of optimising the graphical representations by means of PCoA. In this way, from a population of 36 Canary Islands varieties, a population of 20 was obtained. It should be pointed out once again that under these new circumstances, the population of La Gomera local grapevines was reduced to three individuals (Barrerita negra, Malvasia periquin gomerae, and Verdello gomerae). An assignment test was carried out to check the reliability of the location of each pure individual to its corresponding population. This was 90%. Figure 6 shows the two- and three-dimensional PCoA representations. Figure 6(a.1) shows the two-dimensional PCoA representation by population with a goodness of 69.11%, corroborated by Figure 6(a.2), where the matrix of the Fst statistic for each pair of populations is shown. From this information, it can be seen that POP1 and POP5 are the closest populations occupying a single quadrant, the lower-left one. Also close to them is POP2, positioned in the upper-left quadrant, very close to coordinate 2. In the right quadrants and very distant from this group of populations are POP4 in the upper quadrant and POP3 in the lower quadrant, the latter being the most distant of them all. Figure 6(a.3) shows the two-dimensional representation by PCoA, now by individual. In this case, the distribution is practically the same as that resulting from the population plot but with a rotation on both axes. Thus, POP1 and POP5 appear again very close to each other in the upper-right quadrant (POP5 more dispersed in the centre of the intersection between the axes), while POP2 is close to them and distributed in the lower-right quadrant. Again in the opposite quadrants are POP3 in the upper-left quadrant and POP4 in the lower-left quadrant. In a specific grapevine variety analysis (Figure 6(a.3,b)), the position of the variety Bienmesabe tinto, which is significantly distant from the rest of the group, should be highlighted. The varieties Majorera, Barrerita negra, Malvasia periquin gomerae, and Verdello gomerae are also far from the rest (Figure 6b) although to a lesser extent. While the Majorera variety in Figure 6(a.3) (two-dimensional representation by individual (24.37% goodness of fit)) appears indistinguishable from the rest of the varieties, in Figure 6b (three-dimensional representation by individual (33.87% goodness)), it appears distant from the main group. In both figures, the three local varieties of La Gomera Island also appear in separate locations from the group of Canary Islands *vinifera*, with the variety Barrerita negra occupying a position very close to the variety Bienmesabe tinto. Also, the varieties Malvasia periquin gomerae and Verdello gomerae appear in Figure 6b, standing out from the rest and appearing close to the Majorera grapevine variety.

*3.5. Relation of La Gomera Grapevine Population with Respect to the World Population*

In order to complete and give consistency to the hypotheses set out in the Discussion section, the possibility of comparing the population of 5 local vines from La Gomera Island with the collection of 309 individuals of *Vitis vinifera* ssp. *vinifera* from 22 countries of the world from the TECNENOL database [41,42,45–48], always genotyped with the same 20 SSRs, was considered. This comparative study was carried out using two different strategies. In the first, a purely genetic criterion was taken into account, and in the second, a geographical criterion (country of origin of each variety) was also used.

To implement the first criterion, the Structure 2.3 program was used again so that the best distribution for the 314 varieties under study was sought, in this case between one and nine different distributions. Figure S5 shows that the best distribution for the world population analysed was K = 2, so the components were divided into two populations (Figure 7a and Figure S6). POP1 consists of 183 individuals (58%), mostly from Italy, France and Central Europe, Greece, the Balkan Peninsula, and the Eastern Mediterranean. A total of 88% were pure (161 individuals) and 12% were admixed (22 individuals). POP2, on the other hand, is a population mainly of Spanish origin made up of 131 individuals (42%), of which 103 are pure (79%) and 28 are admixed (21%). The detection of admixed individuals is essential for the necessary data standardisation and thus for the continuation of this study. Therefore, the 22 admixed varieties of POP1 and the 28 admixed varieties of POP2 were removed. In total, 50 individuals were eliminated, reducing the study population

to 264 varieties, with a goodness of assignment of 100%. The Canary Islands varietal population was entirely located in POP2 (Figure S6), and of the 36 varieties from the Canary Islands, only 3 were found to be admixed: Malvasia alistanada fina, Albillo criollo, and Coello blanco (new variety from La Gomera Island).

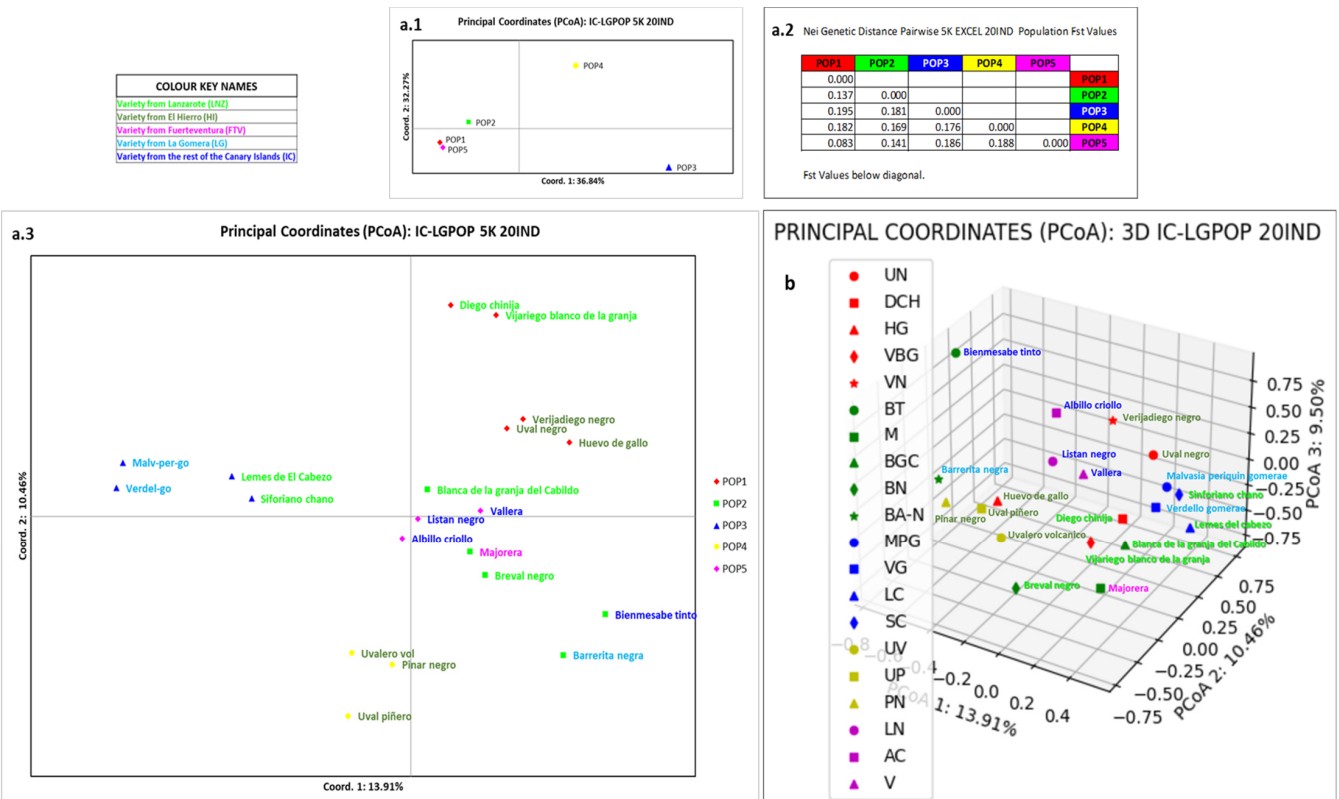

**Figure 6.** Population of varieties of the Canary Islands (20 pure varieties). (**a.1**) Two-dimensional representation of the 5 populations of the Canary Islands by population. (**a.2**) Values of the Fst statistic for each population. (**a.3**) Two-dimensional representation of the 5 populations by individual. (**b**) Three-dimensional representation of the 5 populations by individual.

Figure 7b,c show the circular dendrogram and the phylogenetic tree of the 264 pure individuals distributed in two populations. It can be clearly seen in both cases how the population of varieties from La Gomera Island occupies a branch by itself (either the branch or the variety name coloured in pink). These are the new varieties Malvasia periquin gomerae and Verdello gomerae as well as the already-known Albillo forastero. The Barrerita negra variety is found together with the rest of the Canary Islands varieties in the continuous branch. It should be remembered that the Coello blanca variety was eliminated as it was an admixed variety. Therefore, (a) the first main branch contains the La Gomera Island population, (b) the second main branch is made up of varieties from the Canary Archipelago, where the varieties from El Hierro Island predominate as well as the variety from Fuerteventura Island known as Majorera, and (c) in one of the sub-branches of the third main branch (origin of the Spanish varieties and also of the rest of the world), there is the other half of the Canary Islands varieties, dominated by the varieties from the island of Lanzarote, together with the interspecific cross Bienmesabe tinto.

The two- and three-dimensional PCoA representations are shown in Figure 8. In order to see the extent of the uniqueness of La Gomera and the Canary Islands, both populations were extracted from POP2 and studied as independent populations.

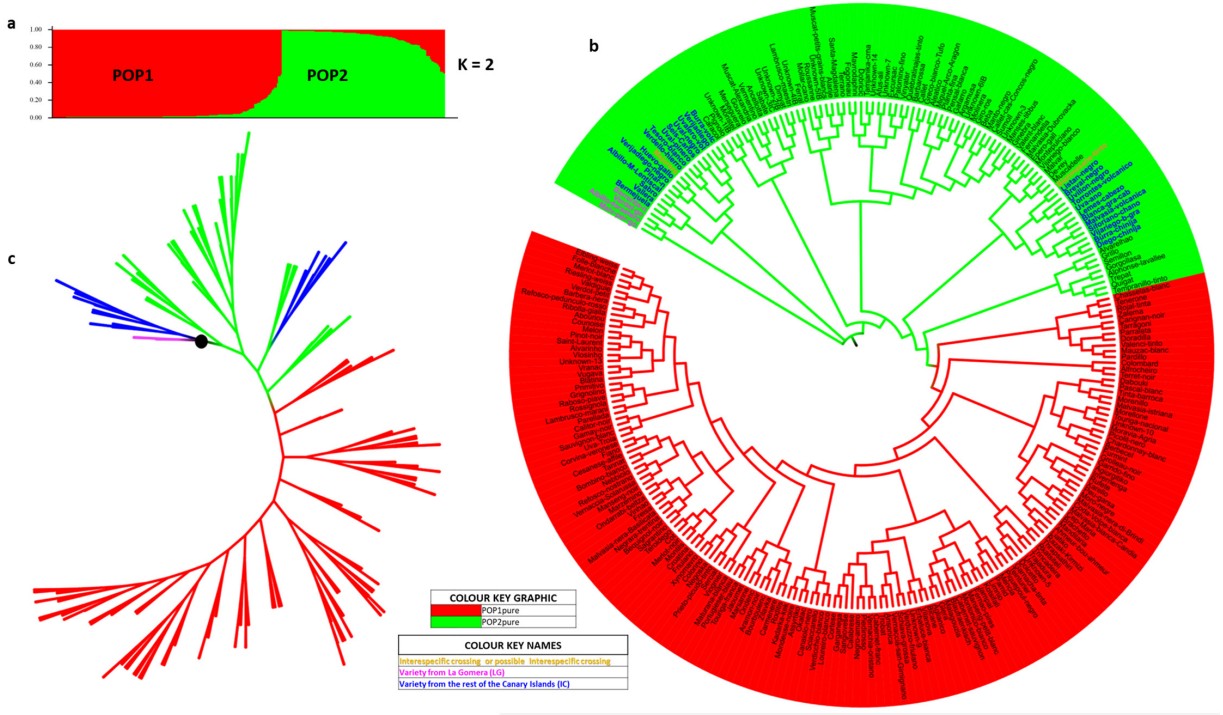

**Figure 7.** World population (314 individuals) distributed in 2 populations. (**a**) Graphical representation of K = 2 according to Structure 2.3 (with pure and admixed individuals). (**b**) Circular neighbour-joining dendrogram of the world population's 264 pure individuals. (**c**) Pure individuals' world population phylogenetic tree.

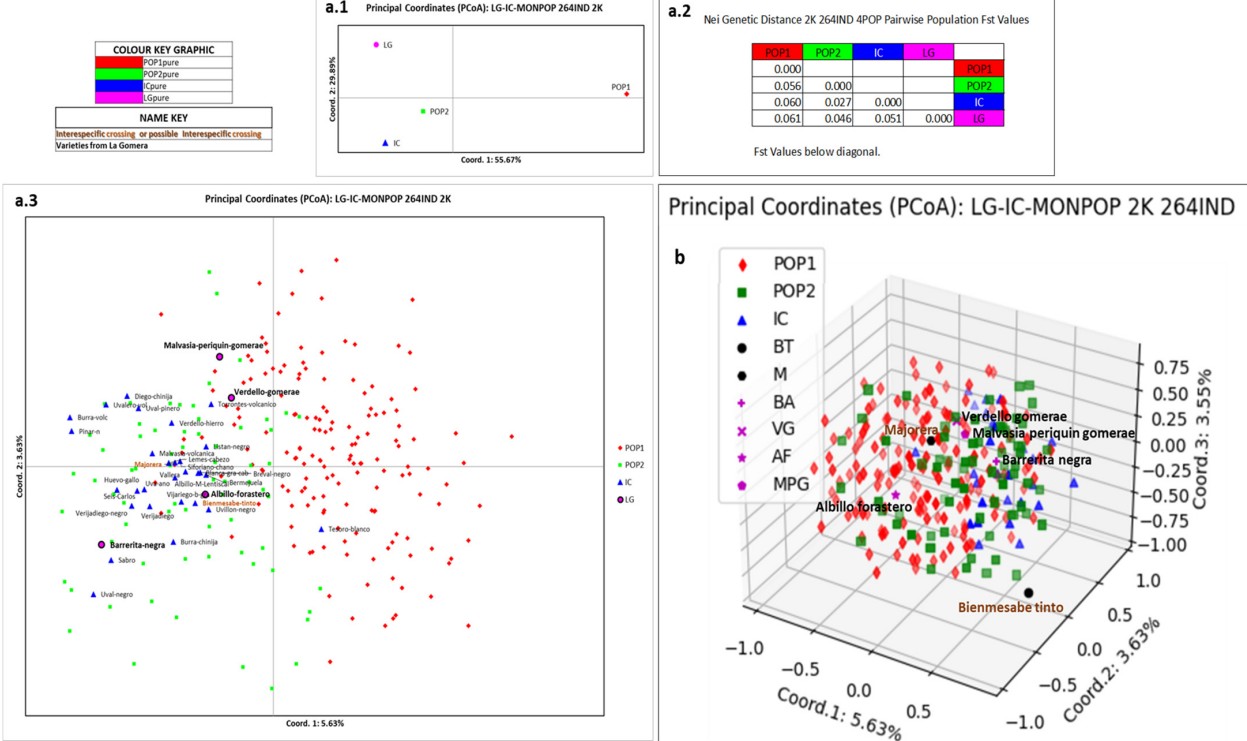

**Figure 8.** PCoA representation of La Gomera, Canary Islands, and worldwide populations normalised for K = 2. (**a.1**) Two-dimensional representation of the 4 populations per population. (**a.2**) Values of the Fst statistic for each population. (**a.3**) Two-dimensional representation of the 4 populations per individual. (**b**) Three-dimensional representation of the 4 populations per individual.

In Figure 8(a.1,a.2) (with a goodness of fit of 85.56%), it can be clearly seen how the La Gomera Island population, occupying the upper-right quadrant, is very distant from both POP2 and IC, which occupy the lower-left quadrant. In Figure 8(a.3,b), corresponding to the representations by individual, it is possible to see the total overlap of the three populations. Perhaps the most relevant fact is that in Figure 8b, the Bienmesabe tinto variety once again moves away from the vinifera pool, and the Majorera variety is apparently integrated with the rest of vinifera.

Once the geographical component was introduced into this study to consolidate the results obtained so far, the distribution of the 314 varieties was observed in seven geographical areas arbitrarily defined according to the country of origin of each variety published in the VIVC [15,49]. It was decided to distribute by area and not by country, as there were countries with only one representative. In this way, the seven areas that were the object of our study were defined as follows (Figure S7): EASTMED-CAU (Algeria, Cyprus, Georgia, Israel, Lebanon, Tunisia, and Turkey), BP (Bosnia and Herzegovina, Bulgaria, Croatia, Greece, Serbia, Slovenia, and Montenegro), ITA (Italy), FRA-CEU (Austria, France, Germany, Hungary, and Switzerland), IP (Spain and Portugal), IC (Canary Islands), and LG (La Gomera Island). Using this distribution, an assignment test was performed with the GenAlEx 6.5 program, which gave a goodness of fit of 61%. This test helped to locate all those varieties that were misplaced. The EASTMED-CAU population, which accounted for 4% of the total, was composed of 33% pure individuals and 67% admixed individuals. In BP, 9% of the world population was found to be half pure and half admixed. ITA had 23% of the representatives, of which 51% were pure and 49% were admixed. FRA-CE had 19% of the population, with 72% pure and 28% admixed varieties. IP was the largest population with 33% of varieties, of which 65% were pure and 35% were admixed. IC accounted for 10% of the world total, and this population was distributed in 75% pure individuals and 20% admixed individuals. Finally, LG, with the smallest population (2% of the total), presented 40% pure individuals (Malvasia periquin gomerae and Verdello gomerae) and 60% admixed individuals (Albillo forastero, Barrerita negra, and Coello blanca). Once the admixed or misplaced varieties were located, they were eliminated (123 individuals), leaving a starting population of 191 pure or well-placed varieties. Thus, with the populations of well-located individuals, a second allocation test was carried out, which showed a goodness of fit of 91%.

Figure 9 shows the circular diagram and the corresponding phylogenetic tree for the world population under the geographical criterion. In both images, it can be seen, in a significant way, how LG has a singular and outstanding character with respect to both the IC varieties and the varieties of the rest of the world.

It is also worth noting that the IC population is distributed in three sub-branches (Figure 9b). In the first, the LG and the Majorera varieties are separated and are almost entirely made up of individuals from the island of El Hierro (Figure 9a). The second sub-branch, linked to the previous ones, is made up of individuals from Lanzarote Island, and the third sub-branch, now linked to peninsular individuals and where the Bienmesabe tinto variety is located, is made up of the rest of the varieties from Lanzarote and varieties from the rest of the Canary Islands. Another remarkable fact is the dichotomy between PI varieties. This population is spread over the three main branches of both figures, the main one being the one most linked to IC. Figure 10 shows the results of the graphical representations using PCoA.

Figure 10(a.1,a.2) with a reliability of 76.74% show the final distribution of the seven world populations. LG is shown to be significantly distant from the rest of the populations, occupying the lower-right quadrant. The rest of the populations occupy the left quadrants. On the other hand, IC and IP, which occupy the upper-left quadrant, are close to each other but differ from the rest of the world populations, which are distributed in the lower-left quadrant. Figure 10(a.3) does not give relevant information, as both LG and IC, and the varieties highlighted as being either possible interspecific crosses or as being a true interspecific cross, overlap with IP. Finally, Figure 10b shows that the Bienmesabe tinto variety (a true interspecific cross) is significantly displaced from the viniferas and that

the pure varieties of LG together with Majorera (a possible interspecific cross) are slightly different from the rest of the viniferas.

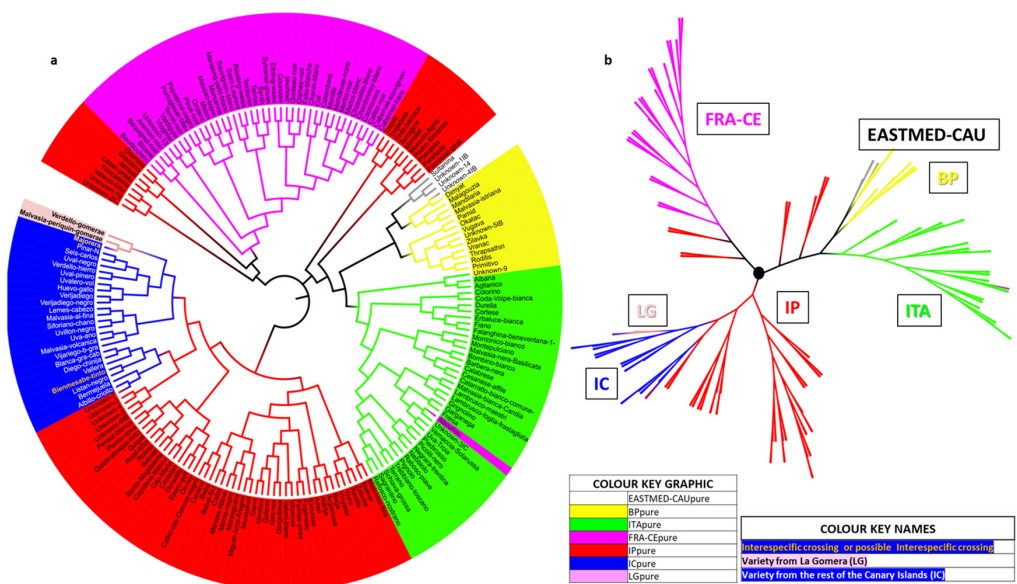

**Figure 9.** World population (191 individuals) distributed in populations corresponding to 7 geographical areas. (**a**) Circular neighbour-joining dendrogram of the 191 pure individuals of the world population, highlighting the Bienmesabe tinto, Majorera, and LG locations. (**b**) Phylogenetic tree of 7 populations' distribution with all their individuals.

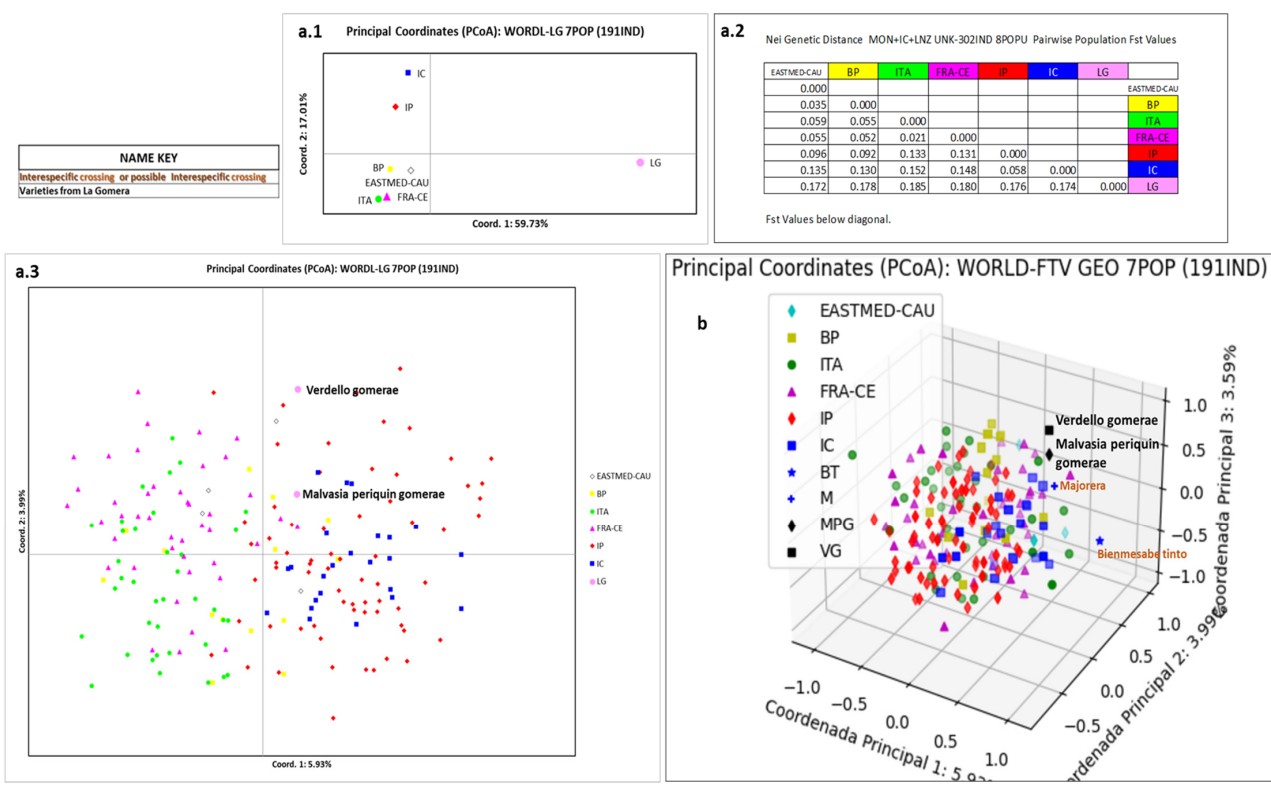

**Figure 10.** PCoA representation of La Gomera, Canary Islands, and world populations for the geographical criterion. (**a.1**) Two-dimensional representation of the 7 populations per population. (**a.2**) Values of the Fst statistic for each population. (**a.3**) Two-dimensional representation of the 7 populations per individual. (**b**) Three-dimensional representation of the 7 populations per individual.

## 4. Discussion

One of the main objectives of this study was to analyse the MP-SSRs of the 120 samples obtained during the La Gomera Island prospection with the 20 SSRs usually analysed by the TECNENOL research group. Once the MP-SSRs were obtained, they were compared with the TECNENOL database [41,42,45–48] and with specialized books on Canary Islands grapevine varieties [14,43,50]. Finally, the MP-SSRs that were unknown were compared with a worldwide database (*Vitis* International Variety Catalogue (VIVC)) [15,49].

The discovery of four new MP-SSRs and an individual with variations concerning one of these new profiles (a mutation), as well as 29 individuals with variations in their MP-SSRs of the most widespread MP-SSR, gives an idea of the degree of evolution of *Vitis vinifera* ssp. *vinifera* on La Gomera Island, also known as Colombina Island. More than 500 years of natural selection, natural crosses, and mutations, as well as anthropogenic selection, have been the driving force behind the adaptation of this subspecies.

As in the rest of the Canary Archipelago, La Gomera was never attacked by *phylloxera*, and for this reason, its adaptive evolution was never interrupted [51,52]. Therefore, one of the objectives of the Discussion section is to justify the uniqueness of the local wine-growing population of La Gomera.

### 4.1. SSR Polymorphism

The first step is to assess the good functionality of the tools used for the analysis, in this case the chosen SSRs. For this purpose, six statistical parameters were used, among which the allelic diversity (Na and He), the goodness of homozygote or F index (i.e., that a homozygote is real and has not lost alleles), and the probability of identity (PI) were estimated (Table S4). A values comparison of the main statistics with respect to the different published works can be a very complex task as there are often variables that are difficult to handle. These are, for example, the number of starting samples, the number and type of SSRs used for the corresponding genotyping, and the uniformity of the population studied, which always depends on the final objective of each study [53]. Because of this, the reference values from island studies are taken as reference values in order to minimise differences. Moita et al. [54] studied a 34-grapevine population from the Azores archipelago with 14 SSRs; Avramidou et al. [55] worked with 56 accessions from the island of Crete, genotyped with 13 SSRs; and TECNENOL prospected three islands with the same 20 SSRs: Lanzarote with 223 samples analysed (statistical study with 99 unique MP-SSRs) [42], El Hierro with 87 accessions genotyped (statistical study with 46 unique MP-SSRs) [41], and Fuerteventura with 40 samples analysed (statistical study with 15 unique MP-SSRs) [45]. In the La Gomera Island grapevine populations study, 120 samples were genotyped, and the statistical study was carried out to determine the goodness of the chosen kit, with 52 unique MP-SSRs. When the Na and/or He (parameters that indicate allelic diversity) are compared, it can be deduced that as the number of individuals in the population which are not of the same variety increases and the number of SSRs tested also increases, the values found will be higher. On the other hand and in general terms, the higher the number of SSRs, the lower the accumulative PI. Therefore, with lower values in the accumulative PI, the ability of a given SSR kit to distinguish two apparently identical samples will show the high discrimination capacity of the chosen SSR kit. The populations of Crete Island [55] and the population of the Azores archipelago [54] showed higher accumulative PI values than those found in La Gomera Island and the rest of the islands surveyed by TECNENOL. This is probably not due to the malfunctioning of the SSR kit used but to the low number of SSRs tested (only 13 or 14 SSRs). Regarding the F index (probability that a kit presents null alleles), the La Gomera population only presents two SSRs with an index higher than 0.01, a threshold above which homozygotes are considered doubtful [56]. In this sense, this work presents the best values of all the studies consulted. Therefore, and comparing the results found on La Gomera Island with the rest of the islands prospected by the TECNENOL research group, it can be concluded that the SSR kit used continues to give good results and is therefore effective and efficient for this research's purposes, as the main parameters do

not show major differences (except for the island of Fuerteventura, where the low number of members of the population means that many of the parameters show different indices from the usual ones). For La Gomera Island, the best-performing SSRs were VVMD28, VVMD36, VVMD27, and ZAG47, and the worst-performing SSRs were, once again, VVS29, VVS3, and UCH19 (Table S4).

*4.2. Grapevine Variety Analysis*

The population of accessions collected in the La Gomera Island prospection turned out to be fairly uniform, since in a first standardisation it went from 120 individuals (Table S1) to 52 unique MP-SSRs (Table S2). This meant the removal of 68 individuals (56.67%) from the population because they possessed the same genetic profile either to an identity 708 or to a mutation. This reduction was slightly higher than that found on the island of Lanzarote [42], with 55% of individuals discarded in the first data normalisation, which was so far the highest (El Hierro had 52.9% [41] and Fuerteventura had 37.5% [45]). This population standardisation of the island of La Gomera also included the elimination of two sports unique in the world, Malvasia rosada [14,43,48–50] and Bermejuela tinta [14]. It is also interesting to note that these 52 unique MP-SSRs correspond to representatives of 19 varieties, which demonstrates the existence of a very significant intravarietal variability (63%).

Of these 19 varieties, 3 correspond to individuals described for the first time, the varieties Coello blanca, Malvasia periquin gomerae, and Verdello gomerae, and another individual that was described by Rodríguez-Torres in 2013 [43] as unknown no. 2, which in this work is recorded as Barrerita negra. It should be noted that for the variety Verdello gomerae, another sample was also collected which turned out to be a mutation in two alleles of this new variety, registered as Verdello gomerae de Monacal. The consideration of an MP-SSR as corresponding to a new variety is somewhat arbitrary for the scientific community. Different authors established different thresholds, for example, Ibañez et al. [57] (SSR (2 alleles/26), 92%), Vélez [58] (SSR (2 alleles/18), 89%), and Cabezas et al. [59] (SNP, 90%). For this work, the threshold was set at a difference of 5 alleles out of the 40 studied for 20 SSRs, which corresponded to an MP-SSR similarity of 87.5% with respect to the most widespread MP-SSR in the TECNENOL database.

Continuing with Tables S1 and S2, it can be stated that in this study, six errors were detected in the registration of accession names with respect to their MP-SSRs, probably due to a lack of knowledge by the vine grower (in red), and 14 entries registered were identified as unknown individuals (pale green). Also, 29 new MP-SSRs for known varieties (intense green) and six more profiles were recorded that were already published in other TECNENOL studies (Forastera de la Isla Redonda and Bermajuelo del Echedo [41]; Listan negro santanero, Listan negro de la corona, Listan blanca chicharrera, and Listan blanco de la bodega [42]). In addition, cases of tri- and quadruple allelism were detected, probably due to hybridisation or periclinal chimeras. This phenomenon was previously described by other authors [60–62], and individuals showing triallelic SSRs were also found in the Lanzarote Island prospection [42], one of them (Listan blanco de la bodega) coinciding with sample codes 1111 and 3111 from the La Gomera prospection (ZAG83). The samples with triallelic SSRs were Listan negro de lo Machado and Listan blanco de Vallehermoso, also in the SSR ZAG83; Malvasia blanca piedra gorda (SCU06); Listan blanco de espina (ZAG83); and Verdello blanco del Corte (UCH19). Finally, the samples Tempranillo de Vallehermoso and the new variety Malvasia periquin gomerae showed a quadriallelic SSR (SCU06) [60].

In the lexicography section, the inclusion in the VIVC of the PN of the four new varieties (Barrerita negra, Coello blanca, Malvasia periquin gomerae, and Verdello gomerae) is proposed. It is also proposed that the new MP-SSR names corresponding to mutations of known varieties be included, as the VIVC database has been doing. In this way, the Pinot noir variety mutant is known by the proper name Pinot meunier, and in the case of the sports, these would also receive their own name, for example, Garnacha blanca (berry colour mutation) and Garnacha peluda (hairy Garnacha mutation), names that differ from the PN of the variety, which is Garnacha tinta (red Garnacha). Therefore,

29 more names should also be included (Forastera blanca de Agulo, Forastera blanca de Vallehermoso, Forastera blanca roquillos, Forastera blanca simancas, Forastera blanca tamargada, Alicante tintilla, Marmajuelo de Vallehermoso, Marmajuelo de Valle bajo, Listan negro de Hermigua, Listan negro de Vallehermoso, Listan negro de lo Machado, Malvasia blanca de Agulo, Malvasia blanca de Vallehermoso, Malvasia blanca piedra gorda, Negramoll de Vallehermoso, Mulata del macayo, Moscatel de Hermigua, Moscatel de la caleta, Moscatel de la caleta fino, Listan blanco de Hermigua, Listan blanco de Vallehermoso, Listan blanco de espina, Ruby cabernet ingenio, Negra Mora, Tempranillo de Vallehermoso, Torrontes volcanico montoro, Torrontes volcanico machado, Uva de año montoro, and Verdello blanco del Corte), as should the new variety mutation (Verdello gomerae de Monacal) and the new sport PN (Bermejuela negra). The inclusion of three new synonyms is also proposed: Marmajuelo Corto as a synonym of Bermajuelo del Echedo [41], Marmajuelo negro as a synonym of Bermejuela negra, and Moscatel Blanca as a synonym of Moscatel de Hermigua. Finally, there are two synonyms registered for another variety that are proposed to be included in a given variety. This is the case of the term Listan negra, a synonym registered for the variety Listan prieto, now proposed as a synonym of Listan negro, and the case of the name Malvasía blanca, a synonym registered for the variety Alarije, now proposed as a synonym of Malvasía Dubrovacka [49].

### 4.3. Genetic Structure of the Grapevine Population of the Island of La Gomera

The population of varieties found in the La Gomera Island prospection, as mentioned above, was 19 varieties. Four of these were new local varieties from La Gomera: Coello blanca, Barrerita negra, Malvasia periquin gomerae, and Verdello gomerae. The remaining 15 varieties turned out to be well-known varieties (Albillo forastero, Alicante Henri Bouschet, Bermejuela, Caracol, Listan negro, Malvasia Dubrovacka, Mollar cano, Muscat of Alexandria, Palomino fino, Ruby cabernet, Tempranillo tinto, Torrontes volcanico, Uva de año, Verdelho branco, and Verijadiego).

The most relevant conclusions after observing the figures related to this section (Figures 3, 4a,b, S1 and S2) are as follows: (a) The new local varieties Malvasia periquin gomerae and Verdello gomerae have a possible influence from the BP (Eastern Mediterranean). These varieties present a very characteristic and singular MP-SSR since in all the graphical representations by PCoA they appear close to the variety Malvasia Dubrovacka (chlorotype A, existing but not widespread in the BP [63]) but significantly distant from the rest (Figure 4), thus forming a single grouping (POP1) (Figure 3). (b) The MP-SSR of the Coello blanca variety probably has a Central European influence. It appears (Figure 4) also close to the Portuguese variety Verdelho branco (chlorotype D), which is a cross between the Central European variety of unknown origin and chlorotype D, Savagnin blanc, and an unknown variety. It is not uncommon that many varieties from the north of the Iberian Peninsula (Spain and Portugal) have as a progenitor the Savagnin blanc variety, which emigrated from the centre of Europe to establish and proliferate in this area through the Camino de Santiago [64]. Both varieties, Coello blanca and Verdelho branco, also have an MP-SSR far from the rest of the varieties that make up the population of La Gomera, probably due to their Central European origin (Figure 4). (c) The best-known local variety of La Gomera Island is the Albillo forastero. This, alongside the local variety from La Palma Island, Albillo criollo, is the result of a cross between the Spanish variety Palomino fino (chlorotype D) and the Portuguese variety Verdelho branco (chlorotype D) [14,63]. In Figure 4b, it is located between its progenitors but close to the other local Gomeran variety, Coello blanca. (d) The last local variety of La Gomera, known as Barrerita negra, is also singular as it is located equidistant from the previous ones but significantly distant from the group of viniferas. This time, in Figure 4a,b, it is positioned close to the Alicante Henri Bouschet variety (chlorotype A), a cross made by Louis Bouschet in 1829 [49] with the French varieties Teinturier (chlorotype A and a cross between Savagnin blanc (chlorotype D) and an unknown variety) and Aramon noir (chlorotype D and a cross between the French variety Oliven (chlorotype D) and the Central European variety of unknown origin He-

unisch weiss (chlorotype C)). Therefore, it is possible that the MP-SSRs of the local varieties from La Gomera Island are not at all conventional, at least not the new local varieties.

Finally, the reliability rates of 27.82% for the two-dimensional PCoA representation and 36.96% for the three-dimensional PCoA representation (Figure 4) should be noted. The overall performance of the PCoA representations decreases as we increase the number of samples to be represented, sometimes reaching no more than 10% in studies with global populations. This is due to the fact that when working with 20 SSRs, 40 allelic values will define the position of a point (variety) in the PCoA plot. This means that the ideal representation, without error and therefore with 100% goodness of fit, would be a 40-dimensional representation. Given the impossibility of executing graphical representations with so many dimensions, the scientific community has to assume as good the reduction to two or three dimensions and their intrinsic errors, a fact that makes us accept this loss of reliability of the varieties represented by PCoA [65,66]. The solution to this problem is to assume trends, not accuracies.

### 4.4. Relation of La Gomera Grapevine Population with Respect to the Canary Archipelago Population

A comparative study was carried out between the local varieties of La Gomera Island and the local varieties of the rest of the Canary Islands in order to see if the affinities and trends between them provided relevant information. To this end, a population structure study was carried out using Structure 2.3. This study proposed the distribution of the 36 Canary Islands local varieties in five ancestral populations (Figures 5 and S4). Although very little information on crosses or on the chlorotypes of these varieties is available, trends can be observed at first sight in this distribution. Furthermore, once the standardisation was performed by eliminating the admixed individuals, Figure 6 as a whole also provides relevant information.

In this sense, the POP3 population (Figures 5 and S4) is the most remote population (Figure 6(a.1,a.2)); as it is made up of representatives closely related to Malvasia Dubrovacka, it is the most Eastern Mediterranean Basin (BP)-influenced population. This population includes Malvasia volcanica, the best-known local variety on the island of Lanzarote and an offspring of Malvasia Dubrovacka [14,42]. The other parent, the Canary Islands variety Bermejuela, also belongs to this group. The rest of the components of this group, in all the studies carried out by the TECNENOL research group [41,42,45–48], have always been related either to Malvasia Dubrovacka or to oriental varieties. Malvasia periquin gomerae and Verdello gomerae form part of this group as pure and local varieties of La Gomera Island.

The POP4 population, also positioned significantly away from the rest and occupying a single quadrant (Figure 6(a.1,a.2)), is a pure population constituted by El Hierro Island individuals. Nearly all of its components (83.33%) are individuals from El Hierro Island. As it is one of the archipelago's westernmost islands, its relationship with the Azores and Madeira was strong [51,52], so much so that the Verdello de El Hierro variety (admixed) is the result of a cross between the local variety of El Hierro Island, Verijadiego (admixed with POP2), and the Portuguese Alfrocheiro (the result of a cross between the Central European variety Savagnin blanc and an unknown variety) [14]. There is no local variety from the island of La Gomera in this grouping. The POP2 population, positioned in the upper-left quadrant very close to the axis and also very close to POP1 and POP5 (Figure 6(a.1,a.2)), is constituted by a group of very diverse grapevine varieties (Figures 5 and S4). It can be observed in this cluster the variety Bienmesabe tinto, described as an interspecific cross, and also the variety Majorera (a possible interspecific cross) [45] as well as the Gomeran variety Barrerita negra. Other varieties that appear in previous studies close to Bienmesabe tinto are those from Lanzarote Island, Blanca de la granja del Cabildo and Breval negro [45]. Also noteworthy in this group are the variety Tesoro blanco from El Hierro, always admixed, and vinifera but with a very distant origin from the population of the Canary Islands [41]. It is interesting to note how Barrerita negra is always positioned very close to the variety Bienmesabe tinto (Figure 6(a.3,b)), thus allowing us to hypothesise

about a possible interspecific origin of this variety from the island of La Gomera. The POP5 population is a cluster characterised by the varieties that are part of the Spanish variety Palomino fino progeny [14,43,50]. Individuals such as Listan negro (Palomino fino × Mollar cano), Albillo criollo de la Isla de La Palma, and its sister of both parents, Albillo forastero from La Gomera Island (Palomino fino × Verdelho branco (Savagnin blanc × unknown)) [14,43,50]. In this grouping and as an admixed variety, the Gomeran variety Coello blanco also appears. In Figure 4a,b, this variety appears very close to the Portuguese variety Verdelho branco, and therefore it is hypothesised that it might have a Central European origin. Finally, the POP1 population is presented as a population whose components come almost equally from the islands of El Hierro and Lanzarote (Figures 5 and S4). Furthermore, this population occupies the lower-left quadrant, very close to POP5 (Figure 6).

In conclusion, the uniqueness of the new local varieties of La Gomera Island is confirmed for the second time, with the Albillo forastero variety having a more integrated MP-SSR in the vinifera group, i.e., not so unique. It is hypothesised that the MP-SSRs of the Coello blanco grapevine variety have a strong Central European influence, and those of the Malvasia periquin gomerae and Verdello gomerae varieties have a marked influence from BP (Eastern Mediterranean Basin). Finally, it is also hypothesised about the possibility that the Barrerita negra variety comes from an interspecific cross.

### 4.5. Relation of La Gomera Grapevine Population with Respect to the World Population

The next step was to broaden the comparative universe and to see how the new local varieties from La Gomera Island would behave, and in turn, how the La Gomera population as a whole would behave. Also, the opportunity to observe the Bienmesabe tinto and Majorera varieties' behaviour related or possibly related to other species of the genus *Vitis*.

A purely genetic strategy and a genetic–geographical strategy were used. In both strategies, admixed varieties from La Gomera were removed. The one that was always eliminated was the white Coello variety, because it was either admixed or badly located. This variety could have an MP-SSR with Central European influence due to its proximity in Figure 4a,b to the Portuguese variety Verdelho branco (Savagnin blanc × unknown). In addition, in the genetic–geographical strategy, the varieties Barrerita negra and Albillo forastero were also eliminated.

Another fact common to both strategies was the remarkable singularity of the La Gomera Island population in the circular dendrograms, phylogenetic trees, and PCoA graphs by population (Figures 7b,c, 8(a.1,a.2), 9a,b and 10(a.1,a.2)). In all these cases, it is clear how the La Gomera population (a) forms a main branch (Figure 7b,c) when the population is made up of the local varieties Malvasia periquin gomerae, Verdello gomerae, and Albillo forastero, forming part of the genetic strategy; (b) forms a sub-branch (Figure 9a,b) when the population consists of the grapevine varieties Malvasia periquin gomerae and Verdello gomerae; or (c) occupies a quadrant on its own away from the other populations in both the genetic and genetic–geographical strategies when the population consists only of varieties from La Gomera Island, Malvasia periquin gomerae and Verdello gomerae (Figures 8(a.1,a.2) and 10(a.1,a.2)). This may highlight the fact that the uniqueness of LG is given by the MP-SSR of either the two varieties or one of them. With regard to the Bienmesabe tinto and Majorera varieties, it should be noted that in all cases, they appear quite separate but with the common feature that Bienmesabe tinto is more closely related to the varieties from Lanzarote Island, while the Majorera variety appears closely linked to the varieties from El Hierro Island.

In the case of the three-dimensional PCoA by individuals' representations, in Figure 8b, it is apparently only the variety Bienmesabe tinto that moves away from the whole, while in Figure 10b, both the varieties Bienmesabe tinto and Majorera as well as the two local varieties of LG that make up this population (Malvasia periquin gomerae and Verdello gomerae) are significantly separated from the rest.

Finally, another relevant fact related to IP should be analysed. In Figure 9a,b, IP presents a fragmented population in such a way that a couple of IP varieties' sub-branches are in the main branch that gives rise to the FRA-CEU population, and another couple of IP sub-branches are located in another of the three main branches, the one that gives rise to the EASTMED-CAU, BP, and ITA varieties. It is clear that the Iberian Peninsula over the years had major stages in its history that were linked to a strong bidirectional transfer, in this case of wood or vine seeds. From the arrival of the Phoenicians and Greeks to the conquest by the Crown of Aragon of the Balkan territories via central-southern Italy [67], the Iberian Peninsula had close bonds with the eastern part of the Mediterranean Basin. This fact could well justify the influence on the MP-SSRs of certain varieties by the EASTMED-CAU, BP, and ITA populations. The previously mentioned case of the Camino de Santiago should also be remembered, which perfectly justifies the Central European influence on the MP-SSRs of the northern varieties of IP [64].

## 5. Conclusions

The conclusions reached after genotyping 120 samples from the prospection of the volcanic island of La Gomera are diverse and interesting. As usual, the kit of 20 SSRs performed efficiently and effectively, although not all of them had the same qualitative level. Four new varieties from La Gomera Island (Coello blanca, Barrerita negra, Malvasia periquin gomerae, and Verdello gomerae) and a mutation of the variety Verdello gomerae, known as Verdello gomerae de Monacal, are presented for the first time and proposed for inclusion in the VIVC. For the first time, 29 unique MP-SSRs corresponding to variations of known varieties were also computed and are also proposed for inclusion in the global database. Additionally, six errors were detected, 14 unknown varieties were identified, and individuals with cases of triallelism and quadriallelism were described.

About the lexicography, four PNs are presented (Coello blanca, Barrerita negra, Malvasia periquin gomerae, and Verdello gomerae), as is one name of a mutation of a new local variety (Verdello gomerae de Monacal), the inclusion of the sport Bermejuela negra, and the registration of twenty-nine names of new mutations corresponding to known varieties. Additionally, three new synonymous names are presented: Marmajuelo Corto as a synonym of Bermajuelo del Echedo; Marmajuelo negro as a synonym of Bermejuela negra (PN of the sport); and Moscatel Blanca as a synonym of Moscatel de Hermigua. Finally, there are two synonyms registered for another variety that we propose be included in a given variety. This is the case of Listan negra, a synonym registered for the variety Listan prieto, now proposed as a synonym of Listan negro, and Malvasía blanca, a synonym registered for the variety Alarije, now proposed as a synonym of Malvasía Dubrovacka.

The population of local grapevines from La Gomera is thought to be the most unique in the Canary Islands to date. It is hypothesised that varieties with a strong influence from the Eastern Mediterranean Basin, Malvasia periquin gomerae and Verdello gomerae, are possibly the most unique. The white Coello variety, which is admixed, seems to have a strong Central European influence. It is hypothesised that the black Barrerita variety as well as the Fuerteventura Majorera variety may be the result of interspecific crossbreeding, as the Bienmesabe tinto variety turned out to be.

**Supplementary Materials:** The following supporting information can be downloaded at: https://www.mdpi.com/article/10.3390/horticulturae10010014/s1, Table S1: Information on 120 accessions from La Gomera (original and conclusive) [15,41–43,45–49]; Table S2: Unique molecular profile of the 52 accessions collected in La Gomera Island prospection. International SSRs coinciding with the TECNENOL SSRs [15,41–43,45–49]; Table S3: Annealing temperature (Ta), range (in base pairs) where the peaks have to appear in the electropherogram and type of fluorescence label of the 20 SSRs used by TECNENOL; Table S4: Characteristics of the twenty microsatellite markers used in this study; Figure S1: The four steps of the graphical method of Evanno et al. [40], allowing the estimation of the true number of ancestral K groups for a population with 19 individuals from La Gomera Island; Figure S2: Genetic structure of the La Gomera Island population. Distribution K = 2 (individuals belonging to each group or population). Details of the ratio of pure and admixed

individuals according to the value of $q$ (pure ($q \geq 85\%$) and admixed ($q < 85\%$)); Figure S3: The four steps of the graphical method of Evanno et al. [40], allowing the estimation of the true number of ancestral K groups for a population with 36 individuals from Canary Islands collection (IC including La Gomera Island); Figure S4: Genetic structure of the Canary Islands population (36 varieties). Distribution K = 5 (individuals belonging to each group or population). Details of the ratio of pure and admixed individuals according to the value of $q$ (pure ($q \geq 85\%$) and admixed ($q < 85\%$)); Figure S5: The four steps of the graphical method of Evanno et al. [40], allowing the estimation of the true number of ancestral K groups for a population with 314 individuals from the TECNENOL database (including La Gomera Island); Figure S6: Genetic structure of the world population. Distribution K = 2 (individuals belonging to each group or population). Details of the proportion of pure and admixed individuals as a function of $q$ value. Nationalities that make up pure and admixed groups; Figure S7. Genetic structure of the world population (191 individuals). Distribution in 7 geographical areas. Detail of the ratio of well-assigned (pure) and misassigned (admixed) individuals. Nationalities that make up each of the groups: EASTMED-CAU (Algeria, Cyprus, Georgia, Israel, Lebanon, Tunisia, and Turkey), BP (Bosnia and Herzegovina, Bulgaria, Croatia, Greece, Serbia, Slovenia, and Montenegro), ITA (Italy), FRA-CEU (Austria, France, Germany, Hungary, and Switzerland), IP (Spain and Portugal), IC (Canary Archipelago), and LG (La Gomera Island).

**Author Contributions:** Conceptualization, F.F. and Q.L.-Y.; methodology, F.F., Q.L.-Y. and C.V.; software, P.S.-G., J.M.C. and F.Z.; validation, F.F., Q.L.-Y. and C.V.; formal analysis, Q.L.-Y. and C.V.; investigation, F.F. and Q.L.-Y.; resources, F.F. and Q.L.-Y.; data curation, F.F., Q.L.-Y. and C.V.; writing—original draft preparation, F.F., Q.L.-Y. and P.S.-G.; writing—review and editing, F.F. and P.S.-G.; visualization, P.S.-G., J.M.C. and F.Z.; supervision, F.F., Q.L.-Y., C.V., P.S.-G., J.M.C. and F.Z. All authors have read and agreed to the published version of the manuscript.

**Funding:** This project has been financed by the Cabildo Insular de isla de La Gomera through the Consejo Regulador de la Denominación de Origen Vinos de La Gomera.

**Data Availability Statement:** Data are contained within the article and Supplementary Materials.

**Acknowledgments:** The authors are grateful to Javier Ibañez and the viticulture specialists Armenia Mendoza and Nancy Melo for their valuable support. We would also like to thank Núria Boronat, Rosa Pastor, Isabel Araque, Braulio Esteve-Zarzoso, Laia Fañanás, and Santiago Moreno for their support in the laboratory.

**Conflicts of Interest:** The authors declare no conflict of interest.

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
