# Peer review of "Analysis of the Diversity Presented by Vitis vinifera L. in the Volcanic Island of La Gomera (Canary Archipelago, Spain) Using Simple Sequence Repeats (SSRs) as Molecular Markers"

_horticulturae, doi:10.3390/horticulturae10010014_

Round 1
Reviewer 1 Report
Comments and Suggestions for Authors
The manuscript is of interest characterizing the vitis vinifera varieties in La Gomera Island. Nevertheless it is somehow confusing to follow the results when it were included all the results with the world population. I suggest that this data also goes to supplementary data, so the manuscript could focus only on the data from La Gomera. The results with the synonyms and homonymes found could be presented in a Table, so it would be easier to retain. Please pay attention to the correction of the names of the varieties. For instance IT IS MISWRITTEN the name of the variery Savagnin. It is writen by mistake Savaning, several times through the manuscript.
Author Response
Reviewer 1
(x) I would not like to sign my review report
( ) I would like to sign my review report
Quality of English Language
( ) I am not qualified to assess the quality of English in this paper
( ) English very difficult to understand/incomprehensible
( ) Extensive editing of English language required
( ) Moderate editing of English language required
( ) Minor editing of English language required
(x) English language fine. No issues detected
|
Yes |
Can be improved |
Must be improved |
Not applicable |
|
|
Does the introduction provide sufficient background and include all relevant references? |
(x) |
( ) |
( ) |
( ) |
|
Are all the cited references relevant to the research? |
( ) |
(x) |
( ) |
( ) |
|
Is the research design appropriate? |
(x) |
( ) |
( ) |
( ) |
|
Are the methods adequately described? |
(x) |
( ) |
( ) |
( ) |
|
Are the results clearly presented? |
( ) |
(x) |
( ) |
( ) |
|
Are the conclusions supported by the results? |
(x) |
( ) |
( ) |
( ) |
Comments and Suggestions for Authors
First of all, the authors would like to thank the full revision of the manuscript by the reviewer. The suggestions and comments have been really useful for the improvement of the paper. Please, find attached our point-by-point responses to each of the comments and suggestions.
In the corrected Manuscript the suggested changes will be highlighted in green.
Thank you very much for taking your time to improve our manuscript.
The manuscript is of interest characterizing the vitis vinifera varieties in La Gomera Island. Nevertheless it is somehow confusing to follow the results when it were included all the results with the world population. I suggest that this data also goes to supplementary data, so the manuscript could focus only on the data from La Gomera.
We humbly have to tell reviewer number one that, more or less, we have followed the same "modus operandi" and manuscript structure as in the last 5 of the 6 articles that this research group has on this topic published in 2 different journals.
- Marsal, G.; Mateo, J.M.; Canals, J.M.; Zamora, F.; Fort, F. SSR analysis of 338 accessions planted in Penedes (Spain) reveals 28 unreported molecular profiles of Vitis vinifera Am. J. Enol. Vitic. 2016, 67, 466-470. https://doi.org/10.5344/ajev.2016.16013
- Marsal, G.; Bota. J.; Martorell, A.; Canals, J.M.; Zamora, F.; Fort, F. Local cultivars of Vitis vinifera in Spanish Islands: Balearic Archipelago. Sci. Hortic. 2017, 226, 122–132. https://doi.org/10.1016/j.scienta.2017.08.021
- Marsal, G.; Mendez, J.J.; Mateo-Sanz, J.M.; Ferrer, S.; Canals, J.M.; Zamora, F.; Fort, F. Molecular characterization of Vitis vinifera local cultivars from volcanic areas (the Canary Islands and Madeira) using SSR markers. Oeno One 2019, 4, 667-680. https://doi.org/10.20870/oeno-one.2019.53.4.2404
- Fort, F.; Marsal, G.; Mateo-Sanz, J.M.; Pena, V.; Canals, J.M.; Zamora, F. Molecular characterisation of the current cultivars of Vitis vinifera in Lanzarote (Canary Islands, Spain) reveals nine individuals which correspond to eight new varieties and two new sports. Oeno One 2022, 56, 281-295. https://doi.org/10.20870/oeno-one.2022.56.3.5519
- Fort, F.; Lin-Yang, Q.; Suárez-Abreu, L.R.; Sancho-Galán, P.; Canals, J.M.; Zamora, F. Study of Molecular Biodiversity and Population Structure of Vitis vinifera ssp. vinifera on the Volcanic Island of El Hierro (Canary Islands, Spain) by Using Microsatellite Markers. Horticulturae 2023, 9, 1297. https://doi.org/10.3390/horticulturae9121297
- Fort, F.; Lin-Yang, Q.; Valls, C.; Sancho-Galán, P.; Canals, J.M.; Zamora, F. Characterisation and Identification of Vines from Fuerteventura (Canary Volcanic Archipelago (Spain)) Using Simple Sequence Repeat Markers. Horticulturae 2023, 9, 1301. https://doi.org/10.3390/horticulturae9121301
For this reason, we ask you to accept the manuscript in the present form in this regard.
The results with the synonyms and homonymes found could be presented in a Table, so it would be easier to retain.
We humbly have to tell reviewer number one that, more or less, we have followed the same "modus operandi" and manuscript structure as in the last 5 of the 6 articles that this research group has on this topic published in 2 different journals.
For this reason, we ask you to accept the manuscript in the present form in this regard.
Please pay attention to the correction of the names of the varieties. For instance IT IS MISWRITTEN the name of the variery Savagnin. It is writen by mistake Savaning, several times through the manuscript.
We have corrected this notable error throughout the whole manuscript.

Reviewer 2 Report
Comments and Suggestions for Authors
This study aims at describing the genetic diversity and population structure of grape varieties from La Gomera island and to compare it with other grapes from the Spanish archipelago. The number of individuals and markers utilised is adequate; some analyses require additional results and clarifications. A critical aspect is that the paper is difficult to read because some parts are too long or too descriptive, and this affects a global comprehension of the main results achieved.
The introduction is mostly focused on the description of the island from which the vines come from, on the techniques of cultivation and but lacks of important references to genetic studies aiming at assessing the genetic diversity in vines in general and in Spanish landraces specifically. In addition, all the “theoretical” part dealing with taxonomic rules of vine names are not necessary: in this case, the authors should limit to report the problems existing with the names of varieties and possible ambiguities.
Regarding the samples utilised, the author should state clearly in the text what grape varieties and how many are from La Gomera and from elsewhere. Sometimes it is difficult to track this info throughout the manuscript.
The data analysis paragraph must be rewritten explaining the rationale behind each analysis and not listing all the software and explaining “in a robotic way” what they do. Careful attention must be paid to use of grammar and punctuation.
The plots of the first STRUCTURE analysis (the “mean Ln” in particular, Figure S1) indicates that chains did not reach convergence (standard deviation is too high) for runs > K2; this is the reason why K2 is outputted as the best one. The authors must run STRUCTURE changing the burn-in or increasing the number of generations to reduce the standard deviation and allow a proper estimation of K. Indeed, more groups than two are visible in the PCoA.
Some genetic diversity indices are barely discussed (F and He/Ho). Please discuss them also in the context of published results on other grapes.
I also recommend, in the Result section, to split the results of genetic diversity per locus from genetic diversity estimates per “population sample” or variety. In the way it is presented, such part is too schematic and difficult to read. This section is rather long and the way it is presented is not of help. I recommend to cut some unnecessary parts and to make the read more fluent.
Similarly, the Discussion section is too “exhaustive” and sometimes introduce concepts as “mutations” or “triallelism” that are barely discussed and hamper once again the comprehension of the manuscript.
Figure and tables need revision too; sometimes titles or captions are not explicative and there is a lack of connection between analyses and in-text figures in some cases.
For all these reasons, I recommend major revisions and I provide a list of things to fix to help in the reviewing process.
Detailed comments
L15 = what does the author mean for “to measure the evolution of grapevine after more than 500 years (inter- and in-travarietal variability)”? Microsatellites are a powerful tool to assess genetic diversity and “population” structure, not for dating events or inferring evolutionary histories
L18-19 = it is not clear at this stage what are these “unique profiles”, as well as triallelism and quadriallelism. Please revise.
L31 = most relevant to what/whom?
L33 = I am not sure about the meaning of this sentence. Is there another way to produce “wines” without vines?
L44 = the authors could also introduce the need of looking for genotypes (rather than individuals) possessing characteristics as resistance to parasites and drought/ soil salinity
L47-51 = please try to make the paragraph smoother avoiding “bullet sentences” regarding geographic location
L56 = unless there are naturalized vines, it is incorrect to refer to cultivated plants as “flora”
L69 = I find no connection between this statement and the previous one. If the phylloxera plague started at the end of XIX century, why the species has been evolving for 500 years? In addition, a parenthesis is missing, but please avoid parentheses in parentheses.
L110 = is this statement really needed?
L138 = there is a verb missing. Please check the sentence.
L143 = please specify to how many “varieties” these 120 shoots belong
L148 = samples were stored at -20°C (and therefore -200 is a typo) or in a tank containing liquid nitrogen (and the temperature is correct)?
L156 = the Nanodrop provides estimates of concentration, but quality (meant as integrity and not only purity) should be assessed with other methods, e.g., electrophoresis
L165 = maybe “international standards”? In addition, please specify better that Table S2 includes the genotypes of accessions with only those seven loci. This table must be also modified following the suggestions for Table S1 (see below)
L170 = please also report the final volume of each reaction and dNTPs concentration
L189 = If the input files for running the several pop gen software were prepared using GenAlEx, please start the sentence with this info
L190-196 = please reformulate this short paragraph. First, check the use of punctation and parentheses; second, avoid explanations like “this is…”: most of them are not needed.
L201 = you cannot start a sentence with “calculate”
L216 = why 7 and 9 clusters were chosen? According to names listed in supplementary tables, the number of varieties is far bigger (later in the results I see that 19 varieties were left). Please justify your choice and explain how it could have affected the detection of clusters in your accessions.
L218 = please do not start a sentence with “And”
L226-231 = this paragraph is more for discussion than results section. Please move it or remove it
L238 = is not a population, but a population sample, or better analysed accessions
L270- = what are “16 mutations in one allele, 3 mutations on two alleles…”? Are the authors referring to the number of repeats per locus differentiating the analysed accessions from the reference genome? If so, it is not clear in the way it is presented. In addition, the use of expressions as “number of mutations” is more suitable to sequences, not size variants. I suggest to reformulate the entire paragraph that is very difficult to follow.
L288 = any reference to a figure?
L359 = please avoid full stops in titles
L628 = again, it is not clear to me the definition of “mutation” in this context
L636 = “this section”?
L748 = what is a “population of varieties”?
L778 = why talking about chlorotypes when using nuclear markers here? I may have missed this point
L808 = please replace “y” with “and”. Please also do not start sentences with POP
Figure 1. Please add a scale bar for distance. In addition, if La Gomera Island belongs to the Canary islands, why are also Azores, Cape Verde and Madeira highlighted in the left panel?
Figure 2. Please ensure that the quality of figure is high. In addition, the citation in figure legend refers to the fact that it comes from that paper? If so, please ensure you have the permission to publish it.
Figure 4 = the 3D PCoA is difficult to read and the legend overlaps with the figure itself. Please correct this issue.
Figure 7 = names in the tree are not visible and the quality of the figure seems not high
Figure 8 = in panel a.2, please check the title “Nei Geneti 2K (6) ..”. It is not clear.
Table S1 = in the column “% Similarity to the nearest genome” a numeric value is expected, not a text; please correct. Column “c” refers to colour of wine? If so, please add this word. What is the nearest genome? If the reference genome is not always the same, please include the info of reference genome for each entry. The field “different alleles” has only 4 columns: this means that new sizes were only reported for a maximum of four loci? Please explain it better.
Table S3 = The title should also report that the table includes size range (not rang) information and type of fluorescence label
Table S4 = in the title, please replace “characterization” with “characteristics”
Comments on the Quality of English LanguageI strongly recommend to check the English grammar (use of punctuation, parentheses) and pay particular attention to spot typos and Spanish “tracks”. Some sentences start with "and" and this should be avoided.
Author Response
Reviewer 2
(x) I would not like to sign my review report
( ) I would like to sign my review report
Quality of English Language
( ) I am not qualified to assess the quality of English in this paper
( ) English very difficult to understand/incomprehensible
( ) Extensive editing of English language required
(x) Moderate editing of English language required
( ) Minor editing of English language required
( ) English language fine. No issues detected
|
Yes |
Can be improved |
Must be improved |
Not applicable |
|
|
Does the introduction provide sufficient background and include all relevant references? |
( ) |
( ) |
(x) |
( ) |
|
Are all the cited references relevant to the research? |
(x) |
( ) |
( ) |
( ) |
|
Is the research design appropriate? |
( ) |
(x) |
( ) |
( ) |
|
Are the methods adequately described? |
( ) |
( ) |
(x) |
( ) |
|
Are the results clearly presented? |
( ) |
( ) |
(x) |
( ) |
|
Are the conclusions supported by the results? |
( ) |
( ) |
(x) |
( ) |
Comments and Suggestions for Authors
This study aims at describing the genetic diversity and population structure of grape varieties from La Gomera island and to compare it with other grapes from the Spanish archipelago. The number of individuals and markers utilised is adequate; some analyses require additional results and clarifications. A critical aspect is that the paper is difficult to read because some parts are too long or too descriptive, and this affects a global comprehension of the main results achieved.
First of all, the authors would like to thank the full revision of the manuscript by the reviewer. The suggestions and comments have been really useful for the improvement of the paper. Please, find attached our point-by-point responses to each of the comments and suggestions.
In the corrected Manuscript (new version) the suggested changes will be highlighted in green.
Thank you very much for taking your time to improve our manuscript.
The introduction is mostly focused on the description of the island from which the vines come from, on the techniques of cultivation and but lacks of important references to genetic studies aiming at assessing the genetic diversity in vines in general and in Spanish landraces specifically. In addition, all the “theoretical” part dealing with taxonomic rules of vine names are not necessary: in this case, the authors should limit to report the problems existing with the names of varieties and possible ambiguities.
It has been known for decades that both Spain and Portugal are countries where the biodiversity of Vitis vinifera L. is remarkable due to the different studies published and despite the fact that they suffered the phylloxera plague.
However, the authors of this study wanted to give a different approach to the Introduction section, addressing aspects that were relevant to the varieties found on the unknown island of La Gomera and to its economy. These aspects would focus on:
- A contribution to the varietal biodiversity of the vine, presenting the "local" varieties of this island as a tool to stop the homogenisation of the world wine market.
- To be successful in the wine market (apart from having unique varieties in the world), we must obviously have a variety/s that are really good at all organoleptic levels, but we must also have history, viticultural techniques and an environment that is pleasant for the consumer. This is what is known as the "marketing of the senses". It is also important to point out that the winegrowers of the island of La Gomera carry out Heroic Viticulture (in all its aspects).
- Our research group has been studying Canary Islands grapevine varieties for years and in an article we published in 2019 in the journal Oeno One, we proposed to the scientific community to change the main name of the most important variety for the island of La Gomera. This variety, in the Vitis International Variety Catalogue (VIVC) is called Albillo forastero, a name that was heard for the first time by the inhabitants of the Canary Islands archipelago as well as by the citizens of the island of La Gomera. Paradoxically, they had no knowledge of this term. They know this variety as either Forastera blanca or Forastera gomera, or Forastera gomerae (which is the name we propose as the main name). After all these years, in this article we are once again calling for the name change that was not possible in 2019, now in a specific study of the island. It is for this reason that we believe it is important to remember all the information (definitions, comments...) on the terms used, as according to the CURRENT definition of "prime name" of the VIVC itself, the term ALBILLO FORASTERO would have to disappear, in order to call this variety with the best known name on the island of La Gomera and in the Canary Archipelago. The whole information can be seen by reviewer two in: Marsal, G.; Mendez, J.J.; Mateo-Sanz, J.M.; Ferrer, S.; Canals, J.M.; Zamora, F.; Fort, F. Molecular characterization of Vitis vinifera L. local cultivars from volcanic areas (the Canary Islands and Madeira) using SSR markers. Oeno One 2019, 4, 667-680. https://doi.org/10.20870/oeno-one.2019.53.4.2404
Furthermore, this strategy has been used and accepted in our 6 publications, but especially in the last 3 that refer to other islands:
- Marsal, G.; Mateo, J.M.; Canals, J.M.; Zamora, F.; Fort, F. SSR analysis of 338 accessions planted in Penedes (Spain) reveals 28 unreported molecular profiles of Vitis vinifera Am. J. Enol. Vitic. 2016, 67, 466-470. https://doi.org/10.5344/ajev.2016.16013
- Marsal, G.; Bota. J.; Martorell, A.; Canals, J.M.; Zamora, F.; Fort, F. Local cultivars of Vitis vinifera in Spanish Islands: Balearic Archipelago. Sci. Hortic. 2017, 226, 122–132. https://doi.org/10.1016/j.scienta.2017.08.021
- Marsal, G.; Mendez, J.J.; Mateo-Sanz, J.M.; Ferrer, S.; Canals, J.M.; Zamora, F.; Fort, F. Molecular characterization of Vitis vinifera local cultivars from volcanic areas (the Canary Islands and Madeira) using SSR markers. Oeno One 2019, 4, 667-680. https://doi.org/10.20870/oeno-one.2019.53.4.2404
- Fort, F.; Marsal, G.; Mateo-Sanz, J.M.; Pena, V.; Canals, J.M.; Zamora, F. Molecular characterisation of the current cultivars of Vitis vinifera in Lanzarote (Canary Islands, Spain) reveals nine individuals which correspond to eight new varieties and two new sports. Oeno One 2022, 56, 281-295. https://doi.org/10.20870/oeno-one.2022.56.3.5519
- Fort, F.; Lin-Yang, Q.; Suárez-Abreu, L.R.; Sancho-Galán, P.; Canals, J.M.; Zamora, F. Study of Molecular Biodiversity and Population Structure of Vitis vinifera ssp. vinifera on the Volcanic Island of El Hierro (Canary Islands, Spain) by Using Microsatellite Markers. Horticulturae 2023, 9, 1297. https://doi.org/10.3390/horticulturae9121297
- Fort, F.; Lin-Yang, Q.; Valls, C.; Sancho-Galán, P.; Canals, J.M.; Zamora, F. Characterisation and Identification of Vines from Fuerteventura (Canary Volcanic Archipelago (Spain)) Using Simple Sequence Repeat Markers. Horticulturae 2023, 9, 1301. https://doi.org/10.3390/horticulturae9121301
For all these reasons, we ask that you accept the approach of the INTRODUCTION section as presented.
Regarding the samples utilised, the author should state clearly in the text what grape varieties and how many are from La Gomera and from elsewhere. Sometimes it is difficult to track this info throughout the manuscript.
In all our articles, this information is located in the Supplementary Material file. In this case in Tables S1 and S2. In any case, if reviewer two thinks it will improve the understanding of our article, we gladly accept your suggestion. This is collected between lines 268-282 of the new version of the Manuscript.
Thank you so much.
Furthermore, the accessions in these tables (Tables S1 and S2) correspond to 19 varieties, of which 5 are local varieties from the Island of La Gomera (Albillo forastero, Barrerita negra, Coello blanca, Malvasia periquin gomerae, Verdello gomerae), also 5 they are local varieties from the rest of the Canary Archipelago (Bermejuela, Listan negro, Torrontes volcanico, Uva de Año, Verijadiego) and the remaining nine correspond to foreign varieties from the Canary Islands: 3 from Spain (Mollar cano, Palomino Fino, Tem-pranillo tinto ), 2 from Portugal (Caracol, Verdelho branco), 1 from France (Alicante Henri Bouschet), 1 from Greece (Muscat of Alexandria), 1 from the Balkan Peninsula (Malvasia Dubrovacka) and 1 from the United States of America (Ruby cabernet). Mu-tations should also be highlighted, whether color (sport) or numerical (Table S1). There is a color mutation that corresponds to a sport widely spread throughout the Canary Islands (Malvasia Dubrovacka rosada), another that corresponds to a sport from the Is-land of La Gomera (Bermejuela negra), and an individual that presents a numerical variation with respect to a variety from the Island of La Gomera (Verdello gomerae de Monacal).
The data analysis paragraph must be rewritten explaining the rationale behind each analysis and not listing all the software and explaining “in a robotic way” what they do. Careful attention must be paid to use of grammar and punctuation.
The authors understand that it refers to the Materials and Methods section.
Our humble opinion is that if we explain the basis of each of the programs used and also the basis of each of the methods used in each program, this section would be very extensive. That is why in all our articles (we have published 6 on this discipline) we use this "modus operandi". If a researcher is interested in a specific method or program, he can always go to consult the original article, or the program's user manual itself.
You can see this section in our 6 publications:
- Marsal, G.; Mateo, J.M.; Canals, J.M.; Zamora, F.; Fort, F. SSR analysis of 338 accessions planted in Penedes (Spain) reveals 28 unreported molecular profiles of Vitis vinifera Am. J. Enol. Vitic. 2016, 67, 466-470. https://doi.org/10.5344/ajev.2016.16013
- Marsal, G.; Bota. J.; Martorell, A.; Canals, J.M.; Zamora, F.; Fort, F. Local cultivars of Vitis vinifera in Spanish Islands: Balearic Archipelago. Sci. Hortic. 2017, 226, 122–132. https://doi.org/10.1016/j.scienta.2017.08.021
- Marsal, G.; Mendez, J.J.; Mateo-Sanz, J.M.; Ferrer, S.; Canals, J.M.; Zamora, F.; Fort, F. Molecular characterization of Vitis vinifera local cultivars from volcanic areas (the Canary Islands and Madeira) using SSR markers. Oeno One 2019, 4, 667-680. https://doi.org/10.20870/oeno-one.2019.53.4.2404
- Fort, F.; Marsal, G.; Mateo-Sanz, J.M.; Pena, V.; Canals, J.M.; Zamora, F. Molecular characterisation of the current cultivars of Vitis vinifera in Lanzarote (Canary Islands, Spain) reveals nine individuals which correspond to eight new varieties and two new sports. Oeno One 2022, 56, 281-295. https://doi.org/10.20870/oeno-one.2022.56.3.5519
- Fort, F.; Lin-Yang, Q.; Suárez-Abreu, L.R.; Sancho-Galán, P.; Canals, J.M.; Zamora, F. Study of Molecular Biodiversity and Population Structure of Vitis vinifera ssp. vinifera on the Volcanic Island of El Hierro (Canary Islands, Spain) by Using Microsatellite Markers. Horticulturae 2023, 9, 1297. https://doi.org/10.3390/horticulturae9121297
- Fort, F.; Lin-Yang, Q.; Valls, C.; Sancho-Galán, P.; Canals, J.M.; Zamora, F. Characterisation and Identification of Vines from Fuerteventura (Canary Volcanic Archipelago (Spain)) Using Simple Sequence Repeat Markers. Horticulturae 2023, 9, 1301. https://doi.org/10.3390/horticulturae9121301
For all these reasons, we ask that you accept the approach of the Data Analysis section as presented.
Thank you so much.
The plots of the first STRUCTURE analysis (the “mean Ln” in particular, Figure S1) indicates that chains did not reach convergence (standard deviation is too high) for runs > K2; this is the reason why K2 is outputted as the best one. The authors must run STRUCTURE changing the burn-in or increasing the number of generations to reduce the standard deviation and allow a proper estimation of K. Indeed, more groups than two are visible in the PCoA.
In all our articles we have always used the same conditions to develop the Structure program. In the first articles, it is true that we used a greater number of K. We reduced this number to the current number, since beyond these values (K=7, K=9), we did not find significantly interesting results for our purposes.
Now and for this study in particular, our humble opinion is that the problem does not lie in changing the burn-in or lengthening the number of K, but in two factors: the low number of individuals to distribute, and the genetic nature of themselves. The message that emerges from the Structure program is that the two varieties of La Gomera and Malvasía Dubrovacka present profiles so different from the rest that their uniqueness is extremely marked.
It is true that in Figure 4a the two-dimensional PCoA program marks 4 groupings of varieties, with a goodness of 27.81%. But it is also true that if we look at Figure 4b, where the three-dimensional representation is shown, this distribution is clearly reduced again to 2 groups, with a goodness of 36.95%. We cannot represent these individuals in all 40 dimensions, so we have to assume an error in the representation.
In the article referring to the Island of Fuerteventura, we had to apply the Structure 2.3 program. for a population of 10 individuals. It could not be done even for K=2, however, the PCoA graph could be represented in both 2 and 3 dimensions. Please observe Figures 5 and 6 of this article from La Gomera, which correspond to all Canarian varieties (including La Gomera), with 36 individuals that are distributed according to K=5 for the Structure 2.3 program, the correspondence with both the two- and three-dimensional PCoA distribution it does not correspond either.
For all these reasons, we ask you to accept our points.
Some genetic diversity indices are barely discussed (F and He/Ho). Please discuss them also in the context of published results on other grapes.
As indicated in the text, the discussion of these parameters is complex, since it is difficult to compare results for different starting conditions. That is why we set out to compare results between islands:
- Moita et al. [54] studied a 34 grapevine population from the Azores archipelago with 14 SSRs
- Avramidou et al. [55] worked with 56 accessions from the island of Crete, geno-typed with 13 SSRs
- TECNENOL has 3 islands prospected with the same 20 SSRs:
- Lanzarote with 99 unique MP-SSR) [42]
- El Hierro with 46 unique MP-SSR) [41]
- Fuerteventura with 15 unique MP-SSR) [45]
Finally, the comparison was carried out globally and above all comparing the results of the islands belonging to the Canary Archipelago, since at least the SSRs used coincided. The Discussion on this matter was exhaustive but global, demonstrating the effectiveness and efficiency of the SSR Kit used, which is ultimately what it was about.
For all these reasons, we ask you to accept our points.
I also recommend, in the Result section, to split the results of genetic diversity per locus from genetic diversity estimates per “population sample” or variety. In the way it is presented, such part is too schematic and difficult to read. This section is rather long and the way it is presented is not of help. I recommend to cut some unnecessary parts and to make the read more fluent.
The autors are deeply sorry, but we do not understand what you are referring to. We will gladly take into account this matter in the following rounds of review if necessary. Despite this, we want to remind you that this article structure that we presented for the La Gomera study has already been presented in other publications and in three different journals. We kindly ask you to accept this format.
Similarly, the Discussion section is too “exhaustive” and sometimes introduce concepts as “mutations” or “triallelism” that are barely discussed and hamper once again the comprehension of the manuscript.
We try to be thorough in our way of doing things, although we recognize that the successes are not always the desired ones. It is for this reason that we have accepted most of your correct suggestions, and for this we thank you for this effort.
As far as trialelism or quadrialelism is concerned, we certainly do not go into depth, because for this study these concepts are simply considered as another type of mutation different from the numerical mutation (different allelic length) and the color mutation (same genetic profile but different phenotype, what is known as sport). We have given it the same treatment as the previous variations (mutations). Obviously, it is redirected to the corresponding bibliographic studies for researchers who want to delve deeper into this topic.
Figure and tables need revision too; sometimes titles or captions are not explicative and there is a lack of connection between analyses and in-text figures in some cases.
We humbly have to tell reviewer number two that, more or less, we have followed the same "modus operandi" and manuscript structure as in the last two research articles the group has on this topic published in this same journal in 2023. In any case, all the Figures and Complementary Material have been reviewed, correcting different aspects, both grammatical, punctuation, and bibliographic.
For this reason, we ask you to accept our reasoning.
For all these reasons, I recommend major revisions and I provide a list of things to fix to help in the reviewing process.
With the help of your suggestions we have improved the manuscript. Thank you so much.
In red you will find the original text of the Manuscript written
The modifications made are in green.
Detailed comments
L15 = what does the author mean for “to measure the evolution of grapevine after more than 500 years (inter- and in-travarietal variability)”? Microsatellites are a powerful tool to assess genetic diversity and “population” structure, not for dating events or inferring evolutionary histories
In order to measure the evolution of grapevine after more than 500 years (inter- and intra-varietal variability), a prospection has been carried out.
Evidently, you are right.
In this sentence, what we were trying to say was that during the years between the colonization of the island and the present (more or less 500 years), the vine varieties incorporated into the ecosystem of the Island of La Gomera have had to adapt to the conditions of the new environment. This adaptation process has led to the appearance of variations in their genome to survive, that is, to be resilient to all stages of the different stresses to which they were subjected. This adaptation process could have caused the current genetic diversity. Diversity that manifests itself both at an inter- and intra-varietal levels and that is part of the evolution of this domesticated species on the island.
We propose the following reformulation of the phrase:
To measure the genetic diversity of the vine after more than 500 years (inter and intravarietal variability) of adaptation to this new environment, a prospection has been carried out.
L18-19 = it is not clear at this stage what are these “unique profiles”, as well as triallelism and quadriallelism. Please revise.
Four new varieties were found (Coello blanca, Barrerita negra, Malvasia periquin gomerae, Verdello gomerae (and her mutation Verdello gomerae de Monacal)). Also, 29 unique molecular profiles corresponding to mutation of known varieties have been presented for the first time and individuals with triallelism and quadriallelism cases have been described.
We propose the following reformulation of the phrase:
A total of 52 unique profiles were found corresponding to 4 new varieties ((Coello blanca, Barrerita negra, Malvasia periquin gomerae, Verdello gomerae), 9 individuals identical to the most widespread profile, and 39 individuals that presented variations (1 corresponding to a mutation of a new variety (Verdello gomerae de Monacal), 38 corresponding to variations of known varieties, some of which included cases of triallelism or quadriallelism).
L31 = most relevant to what/whom?
Nowadays, cultivation of Vitis vinifera L. varieties is one of the oldest and most relevant worldwide. The FAO (Food and Agriculture Organization of the United Nations) affirms that the most economically valued crop is the vine for vinification of medium and high quality wines [1]. Furthermore, the NASS (National Agricultural Statistics Service) reaffirms its importance as the 6th most economically valuable crop in 2021 in the United States [2]. The world production volume of fresh grapes in 2021 was 74.8 million tonnes, of which more than 20% came from Europe. Specifically, and in this order, Italy, Spain and France lead European production by a wide margin, accounting for 79% of total production [3].
As indicated in the following sentences, which are examples that illustrate the relevance of this crop at an economic level (answering the “what”), international organizations such as the FAO or the OIV [3], or others, such as the NASS (answering the “whom”), are those who affirm this relevance.
But perhaps this paragraph would improve if we rephrased the final part as follows:
The global production volume of fresh grapes in 2021, according to the OIV (International Organization of Vine and Wine), was 74.8 million tons, of which more than 20% came from Europe. Specifically, and in this order, Italy, Spain and France lead European production by a wide margin, representing 79% of total production [3].
L33 = I am not sure about the meaning of this sentence. Is there another way to produce “wines” without vines?
The FAO (Food and Agriculture Organization of the United Nations) affirms that the most economically valued crop is the vine for vinification of medium and high quality wines [1].
You ask an intelligent, interesting and controversial question...
Apparently so! but when only the word wine is used, it will refer to the must or fermented grape juice. For the fermentation of juices from other fruits, the fruit of origin will have to be specified. For example, banana wine can be marketed.
To shed some light on this matter, there is an EU regulation, with the force of law, that accepts the use of the word wine for preparations with fruits other than grapes. This is the Regulation:
REGULATION (EU) No 1308/2013 OF THE EUROPEAN PARLIAMENT AND OF THE COUNCIL of 17 December 2013 establishing the common organization of the markets in agricultural products and repealing Regulations (EEC) No 922/72 , (EEC) No 234/79, (EC) No 1037/2001 and (EC) No 1234/2007.
On page 809, annex VII, part 2, paragraph 1. It clearly says:
However, Member States may authorize the use of the word "wine" if:
- a) is accompanied by a fruit name in the form of a compound name, for the marketing of products obtained from the fermentation of fruits other than grapes, or
- b) is part of a composite denomination.
Any possible confusion with products from the wine categories of this annex will be avoided. IN 20.12.2013 Official Journal of the European Union L 347/809.
Additionally, in the dictionary of the RAE (Academy of the Spanish Language) if you search for the word wine you will be able to find the following:
wine.(From lat. vinum):
- m. Alcoholic liquor that is made from the juice of squeezed grapes, and cooked naturally by fermentation.
- m. Juice from other plants or fruits that is cooked and fermented in the same way as grapes.
But we believe that the text of the article makes it clear. We are talking about "the vine for the vinification of medium and high quality wines", therefore, since the term wine does not have a term as an adjective referring to another fruit, it is understood that we are talking about wine from grape berries.
L44 = the authors could also introduce the need of looking for genotypes (rather than individuals) possessing characteristics as resistance to parasites and drought/ soil salinity
It is therefore a matter of looking for individuals of interest that will help to broaden the range of wines offered on the market and also mitigate the effects of climate change.
The objective of this study is to search for individuals (which are nothing more than gene banks in a single plant) that are different either completely (intervarietal variability (new varieties)) or partially (intravarietal variability (new mutations for a described variety)). All interesting individuals that present unique molecular profiles will be studied in an agronomic characterization in the field and if, in addition to being a new variety (for example), one of these genetic profiles is resistant to a disease or pest, the better. It's not going to be dismissed at all. But we will not know this until subsequent study in the field. At the moment in this study, only unique genetic profiles are sought.
A new phrase proposal is:
It is, therefore, about searching for unique and interesting molecular profiles that not only help to expand the range of wines offered in the current market or mitigate the effects of climate change, but also show advantages to other relevant factors, whether biotic. (pests and/or diseases) and/or abiotics (salinity...).
L47-51 = please try to make the paragraph smoother avoiding “bullet sentences” regarding geographic location
A starting point for this purpose is undoubtedly La Gomera Island. This island is located in European overseas territory, forming part of one of the 17 Spanish Autonomous Communities. La Gomera Island belongs to the Canary Islands volcanic archi-pelago, a group of 8 Spanish islands located in the Atlantic Ocean. The Canary Islands are part of a group of volcanic islands known as Macaronesia, close to the Western Sa-hara (Figure 1) [4].
We present a new proposal:
A starting point for this purpose is undoubtedly La Gomera Island. This island is located in European overseas territory (near Western Sahara), it belongs to the volcanic archipelago of the Canary Islands and Macaronesia (a group of Atlantic volcanic islands) (Figure 1) [4].
L56 = unless there are naturalized vines, it is incorrect to refer to cultivated plants as “flora”
It is now widely accepted that the vine had not been part of the Canary Islands flora until its introduction in the 15th century.
You are right. We propose the following paragraph as an alternative to the existing one:
It is now widely accepted that the vine had not been part of the existing crops in the Canary Islands until its introduction in the 15th century.
L69 = I find no connection between this statement and the previous one. If the phylloxera plague started at the end of XIX century, why the species has been evolving for 500 years? In addition, a parenthesis is missing, but please avoid parentheses in parentheses.
Before the Phylloxera plague, the domesticated grapevine consisted of subspecies, Vitis vinifera ssp vinifera. In this way, the Old World vines were the oldest and evolved over time.
Due to this plague that hit the European continent in the 19th century and later spread to the rest of the world, the vast majority of vineyards disappeared. Due to this catastrophe, the sector decided to look for solutions. Many strategies were tried and of all of them, the one that gave the best results was grafting Vitis vinifera ssp vinifera (the European vine) on a rootstock formed by other American species, whether pure or hybrids.
All areas of the world affected by phylloxera had to be replanted with grafted plants, meaning that the majority of the world's vineyards are no more than 125 years old. On the other hand, the areas of the planet that were not affected by this pest, such as the Island of Crete, the Canary Islands, Chile...their vineyards have been multiplying and propagating for hundreds of years by asexual reproduction (stacking and/or layering), resulting in that both intervarietal and intravarietal variability is greater because mutations have been accumulating for more years.
L110 = is this statement really needed?
Historically, grapevine varieties characterisation and identification were based on the description and comparison of morphological characters. The science that allowed these comparative studies of Vitis vinifera L. phenotype to be carried out was ampelography [17].
Our humble opinion is that yes, it is necessary. The reason is that we think it can be useful to improve the understanding of those scientists specializing in other plant species, who read this article to get ideas and compare results. It can also be useful for young scientists beginning their training.
L138 = there is a verb missing. Please check the sentence.
Finally, to carry out a study of population structure, comparing the population of local varieties in La Gomera with those of other populations in the Canary Islands and the rest of the world.
We propose the following paragraph as an alternative to the existing one:
The last objective was to verify the uniqueness of La Gomera grapevine population. For this purpose, a population structure study was carried out, comparing the population of local varieties of La Gomera with those of other populations in the Canary Islands and the rest of the world.
L143 = please specify to how many “varieties” these 120 shoots belong
In order to explore the Vitis vinifera L. biodiversity extent, 120 vine shoot samples were collected in different municipalities of La Gomera Island (Agulo, Alajeró, Hermigua, San Sebastián de La Gomera, Valle Gran Rey and Vallehermoso)...
Until the samples are analyzed, it is impossible to know with certainty the number of varieties that we can find, since a winegrower may think that his variety is "X" and after the analysis it turns out that this variety is "Y". Not all winegrowers have knowledge of ampelography, which is why errors are detected in all our prospecting work. In any case, by consulting Table S1 of the Supplementary Material file the reader can observe the final results in this paragraph prematurely, and in the Results section definitively.
L148 = samples were stored at -20°C (and therefore -200 is a typo) or in a tank containing liquid nitrogen (and the temperature is correct)?
Once harvested, shoots were stored at -200 °C until processing. Table S1 shows detailed information on the analysed grape varieties.
Samples will be stored at -20°C. It's a typo. Corrected in the text of the new version of the article.
L156 = the Nanodrop provides estimates of concentration, but quality (meant as integrity and not only purity) should be assessed with other methods, e.g., electrophoresis
With the help of the Thermo Fisher® Scientific NanoDrop TM 1000 Spectrophotometer, which accurately assesses the nucleic acid purity level and concentration, the quality of each sample was measured.
We propose the following paragraph as an alternative to the existing one:
With the help of the Thermo Fisher® Scientific NanoDrop TM 1000 spectrophotometer and using the electrophoresis technique, the level of purity, integrity and concentration of the nucleic acid was precisely measured.
L165 = maybe “international standards”? In addition, please specify better that Table S2 includes the genotypes of accessions with only those seven loci. This table must be also modified following the suggestions for Table S1 (see below)
In addition, of this “kit” of SSRs used by TECNENOL, 7 SSRs coincide with those accepted as international [31] (Table S2).
We propose the following paragraph as an alternative to the existing one:
Furthermore, in this SSR “kit” used by TECNENOL there are 7 SSRs that coincide with some of the 9 SSRs accepted by the international scientific community [31]. In Table S2 you can see the values of the allelic lengths of this SSRs considered international standards corresponding to the unique MP-SSRs of this grapevine population.
L170 = please also report the final volume of each reaction and dNTPs concentration
For the satellite regions amplification, 4 ng of DNA and 1 μM of each primer were used, with the particularity that the Forward primer (Fw) was labelled with a specific fluorochrome (6-FAM: VVS3, VVMD7, VVMD28, VVMD36, VrZAG47, VrZAG62, VrZAG83, VvUCH11 y VvUCH19; HEX: VVS2, VVS29, VVMD6, VVMD27, VrZAG21, VrZAG79 y VChr19a; NED: VVMD5, VrZAG64, scu06vv, VvUCH12) using the Am-pliTaq DNA Polymerase kit (Applied Biosystems, Foster City, CA, USA). Seven ther-mocycling blocks were performed according to the different annealing temperatures (Ta) (Table S3) and for all of them, the thermocycling conditions were: a) a first stage of 5 min at 95 °C; b) a second stage of 40 cycles: 45 sec at 95 °C during, 30 sec at the corre-sponding Ta, and 1 min 30 sec at 72 °C and c) a third stage of 7 min at 72 °C. The Ap-plied Biosystems 2720 Thermal Cycler was used for this process (Foster City, CA, USA).
We propose the following paragraph as an alternative to the existing one:
For the satellite regions amplification, the AmpliTaq DNA Polymerase kit (Applied Biosystems, Foster City, California, USA) was used with a final reaction volume of 12.5 μl. The amounts of each reagent broken down per well were: 1.25 μl of Buffer, 2 μl of dNTPs, 0.125 μl of deionized formamide, 0.0625 μl of Taq polymerase, and 4 ng of DNA. In addition to 1 μM of each primer (1.25 μl), with the particularity that the Forward primer (Fw) was labeled with a specific fluorochrome (6-FAM: VVS3, VVMD7, VVMD28, VVMD36, VrZAG47, VrZAG62, VrZAG83, VvUCH11 and VvUCH19; HEX: VVS2, VVS29, VVMD6, VVMD27, VrZAG21, VrZAG79 and VChr19a; NED: VVMD5, VrZAG64, scu06vv, VvUCH12). Seven thermocycling blocks were carried out according to the different annealing temperatures (Ta) (Table S3) and for all of them, the thermocycling conditions were: a) a first stage of 5 min at 95 °C; b) a second stage of 40 cycles: 45 sec at 95 °C for, 30 sec at the corresponding Ta, and 1 min 30 sec at 72 °C and c) a third stage of 7 min at 72 °C. The Applied Biosystems 2720 thermal cycler (Foster City, CA, USA) was used for this process.
L189 = If the input files for running the several pop gen software were prepared using GenAlEx, please start the sentence with this info
GenAlEx 6.5. software was used for different purposes [32,33]. First, to evaluate the efficiency and efficacy of the SSR kit used by TECNENOL. For this purpose,.....
We propose the following paragraph as an alternative to the existing one:
GenAlEx 6.5. software was used for different purposes [32,33]. Data input files were prepared according to this program, using the Excel program (Microsoft). Firstly, the efficiency and effectiveness of the SSR kit used by TECNENOL was evaluated. For this purpose,…
L190-196 = please reformulate this short paragraph. First, check the use of punctation and parentheses; second, avoid explanations like “this is…”: most of them are not needed.
For this purpose, the following statistical parameters were measured: Na (number of different alleles), Ne (number of effective alleles. These are the alleles that are trans-mitted to the next generation), Ho (observed heterozygosity. These are the computed heterozygotes), diversity index or He (expected heterozygosity. Estimation of the het-erozygotes that the population under study could have), F (fixation index. Parameter that measures the goodness of homozygotes) and PI (probability of identity. Probability that two molecular profiles (MP-SSR) with the same SSR values are the same variant).
We propose the following paragraph as an alternative to the existing one:
For this purpose, the following statistical parameters were measured: Na (number of different alleles), Ne (number of effective alleles: alleles that are transmitted to the next generation), Ho (observed heterozygosity: the computed heterozygotes), diversity index or He (expected heterozygosity: estimation of the heterozygotes that the population under study could have), F (fixation index: parameter that measures the goodness of homozygotes) and PI (probability of identity: the probability that two MP-SSR with the same SSR values are the same variant).
L201 = you cannot start a sentence with “calculate”
Secondly, it allowed to find all the Molecular Profiles of SSRs (MP-SSRs) that matched each other, to eliminate redundant information. Assignment tests, to check the sample distribution goodness of fit in different populations, were also performed using this program. GenAlEx 6.5. bases this strategy on the allele frequency of each accession. Calculate a logarithmic probability value of this accession for each subpopulation using the allele frequencies of the respective subpopulations and assign an individual to the subpopulation with the highest logarithmic probability value [34].
We propose the following paragraph as an alternative to the existing one:
Secondly, it allowed to find all the Molecular Profiles of SSRs (MP-SSRs) that matched each other, to eliminate redundant information. Assignment tests, to check the sample distribution goodness of fit in different populations, were also performed using this program. GenAlEx 6.5. bases this strategy on the allele frequency of each accession. It also allowed us to calculate a logarithmic probability value of this accession for each subpopulation using the allele frequencies of the respective subpopulations and assign an individual to the subpopulation with the highest logarithmic probability value [34].
L216 = why 7 and 9 clusters were chosen? According to names listed in supplementary tables, the number of varieties is far bigger (later in the results I see that 19 varieties were left). Please justify your choice and explain how it could have affected the detection of clusters in your accessions.
Population structure was tested from K = 1 to K = 7 for the local population of La Gomera Island and for the Canary Island population, and from K = 1 to K = 9 for the world population.
It is an arbitrary choice resulting from experience. We have published 5 scientific articles on this topic, and although in the older articles the tested K values were higher, we gradually reduced them because they always proposed a K of less than 6 as the best distribution of our population for studies with few individuals, and less than 4 for studies with a large number of individuals.If this was the constant, it is not worth trying the Structure program with more K, since we have always worked with 1,000,000 Markov Chain Monte Carlo (MCMC) steps.
L218 = please do not start a sentence with “And”
All with 10 independent replicates, consisting of 1.000.000 Markov Chain Monte Carlo (MCMC) steps...
We propose the following paragraph as an alternative to the existing one:
Ten independent replications were performed, composed of 1.000.000 Markov Chain Monte Carlo (MCMC) steps after discarding the first 100.000.
L226-231 = this paragraph is more for discussion than results section. Please move it or remove it
Paragraph moved to Discussion section.
L238 = is not a population, but a population sample, or better analysed accessions
The total number of alleles in the population was 181, with a mean value of 9.1.
We propose the following paragraph as an alternative to the existing one:
The total number of alleles in the unique MP-SSR population was 181, with a mean value of 9.1.
L270- = what are “16 mutations in one allele, 3 mutations on two alleles…”? Are the authors referring to the number of repeats per locus differentiating the analysed accessions from the reference genome? If so, it is not clear in the way it is presented. In addition, the use of expressions as “number of mutations” is more suitable to sequences, not size variants. I suggest to reformulate the entire paragraph that is very difficult to follow.
In both tables (S1 and S2) it can be seen that the grouping corresponding to the Al-billo forastero variety consists of 46 individuals and is thus the most numerous. Among them there were 26 accessions that were identical to the most widespread MP-SSR in the TECNENOL database (identities), 16 mutations in one allele, 3 mutations in 2 al-leles and 1 mutation in 3 alleles. This resulted in the definition of 10 different MP-SSRs for this variety. One MP-SSR corresponds to the identity that is named with the main name according to the VIVC, which in this case corresponds to Albillo forastero. Six MP-SSRs showing variation in one allele: a) variation in VVS3-1 (numerical change in the SSR VVS3 first allele), known as Forastera de la Isla Redonda, was previously pub-lished by Fort et al. [45]; b) the VVS3-2 variation which is called Forastera blanca de Agulo ; c) the mutation in VVS29-2 which is known as Forastera blanca de Vallhermoso, in addition, there are two mutated profiles in this same allele but with different allelic lengths which are marked with *, but all of them are known by the same term; and d) the variation in UCH12-2 named in La Gomera as Forastera blanca roquillos. There were also 2 individuals showing variation in 2 alleles of the same SSRs (VVS3-1, VVS29-2) but, as in the previous case, a numerical difference in allelic length was de-tected for VVS29-2. Both samples are known as Forastera blanca Simancas. The last pro-file detected for the Albillo forastero variety corresponded to an accession with a varia-tion in 3 alleles (VVS3-1, VVS29-1, VVS29-2) known as Forastera blanca tamargada in La Gomera Island.
We propose the following paragraph as an alternative to the existing one:
In both tables (S1 and S2) it can be seen that the grouping corresponding to the Albillo forastero variety is made up of 46 individuals and is therefore the most numerous. Among them, 26 accessions were identical to the most widespread MP-SSR in the TECNENOL database (identities), 16 individuals presented variations in their allelic length in one allele, 3 accessions showed variations in 2 alleles and 1 sample presented variation in 3 alleles. These results defined 10 different MP-SSRs for this variety. One MP-SSR corresponded to an identity with the most widespread profile in our database. This profile was named with its PN according to the VIVC, which in this case corresponded to the term Albillo forastero. Six MP-SSRs showed allelic length variation in one allele: a) variation in VVS3-1 (numerical change in the first SSR allele VVS3), known as Forastera de la Isla Redonda, was previously published by Fort et al. [45]; b) the VVS3-2 variation that was called Forastera blanca de Agulo; c) the mutation in VVS29-2 which is known as Forastera blanca de Vallehermoso, in addition, there are two mutated profiles in this same allele but with different allelic lengths that are marked with *, but all are known with the same term; and d) the variation in UCH12-2 called in La Gomera as Forastera blanca roquillos. Two individuals showed variation in their allelic length in 2 alleles of the same SSRs (VVS3-1, VVS29-2) but, as in the previous case, a numerical difference in allelic length was detected for VVS29-2. Both specimens are known as Forastera blanca Simancas. The last profile detected for the variety Albillo forastero corresponded to an accession with variation in allelic length in 3 alleles (VVS3-1, VVS29-1, VVS29-2) known on the Island of La Gomera as, Forastera blanca tamargada.
L288 = any reference to a figure?
Referred at the beginning of the section: 3.2. Grapevine Variety Analysis
All the original and conclusive information concerning the 120 accessions can be found in Table S1. In addition, it presents in detail the similarity of the MP-SSR of a given accession with respect to the most widespread MP-SSR according to the TECNENOL database, even specifying which allele presents the variation. With all this information and the possibility of comparing it with the VIVC database, a study was carried out at both the molecular and terminological scales. In Table S2 and for the 52 unique MP-SSRs, the numerical values of the allelic lengths measured for the 7 interna-tional SSRs available to the TECNENOL research group are also presented.
The presentation of results refers to Tables S1 and S2. It is for this reason that no reference is made to a specific Figure.
L359 = please avoid full stops in titles
3.3. La Gomera Island Grapevine Population. Genetic Structure
We propose the following paragraph as an alternative to the existing one:
3.3. Genetic structure of the grapevine population of the Island of La Gomera.
L628 = again, it is not clear to me the definition of “mutation” in this context
The discovery of 4 new MP-SSRs and a mutation of one of them, as well as 29 individuals with variations in their MP-SSR with respect to the most widespread MP-SSR, gives an idea of the evolution extent of Vitis vinifera ssp vinifera in La Gomera Island, also known as Isla Colombina.
We propose the following paragraph as an alternative to the existing one:
The discovery of 4 new MP-SSR and an individual with variations concerning one of these new profiles (mutation), as well as 29 individuals with variations in their MP-SSR for the most widespread MP-SSR, gives an idea of the degree of evolution of Vitis vinifera ssp vinifera on La Gomera Island, also known as Colombina Island.
L636 = “this section”?
The aim of this section, among others, is to justify the uniqueness of the local grapevine population in La Gomera.
We propose the following paragraph as an alternative to the existing one:
Therefore, one of the objectives of the Discussion section is to justify the uniqueness of the local wine-growing population of La Gomera.
L748 = what is a “population of varieties”?
The population of varieties found in La Gomera Island prospection, as mentioned above, was 19 varieties.
We propose the following paragraph as an alternative to the existing one:
From the population of individuals found in the survey of the Island of La Gomera, as previously mentioned, 19 PM-SSR corresponding to 19 varieties were found.
L778 = why talking about chlorotypes when using nuclear markers here? I may have missed this point
In the paragraph between lines 755 and 784 of the original document, or between lines 782 and 811 of the new document in which the proposed changes are shown, reference is certainly made to chlorotypes as a result of the extraordinary study carried out by the Arroyo-Garcia research group in 2006. The authors have used the information in this great article to justify our results, that is, to try to give a convincing explanation for the position of each variety in each of the graphical representations shown in this article. But we have not only used this "strategy", in different parts of the article, and for this same purpose, we have used the pedigree information published by Rodríguez-Torres (2018) or in the VIVC.
L808 = please replace “y” with “and”. Please also do not start sentences with POP
This changes have been taken into account in the new manuscript.
Figure 1. Please add a scale bar for distance. In addition, if La Gomera Island belongs to the Canary islands, why are also Azores, Cape Verde and Madeira highlighted in the left panel?
We are very sorry to inform you that it is not possible for us to add a stopover. The image does not include it. The other archipelagos appear in this image because our intention is for the reader to get to know Macaronesia. The name Macaronesia is given to the group of these Atlantic islands, all of which are of volcanic origin and to which the Canary Archipelago belongs. We believe that this will be information unknown to most of our readers, and therefore, it will be interesting for them to know it. For all these reasons, we ask you to accept this Figure 1 as we present it.
Figure 2. Please ensure that the quality of figure is high. In addition, the citation in figure legend refers to the fact that it comes from that paper? If so, please ensure you have the permission to publish it.
We've done it. Thank you very much for these suggestions.
Figure 4 = the 3D PCoA is difficult to read and the legend overlaps with the figure itself. Please correct this issue.
The Python Data applying Matplotlib strategy program is a programming program. It is extremely difficult to manipulate. We are sorry but we are not able to prevent the legend box from overlapping the graph. Regarding the topic of names, we can only propose a zoom increase. We think that today most of our readers use the document in PDF format that allows these amplifications without problem. Also, if this research article is finally published, figures by itself can be consulted online one by one.
For all these reasons, we ask you to accept this Figure as we present it.
Figure 7 = names in the tree are not visible and the quality of the figure seems not high
It is impossible to put the names on this phylogenetic tree without there being extensive overlap between them, to the point that the reading of the names becomes intelligible. It is about placing 264 names...there are many. For this reason, the authors preferred not to put the names in this image. The resolution is good, enough to be published without problems. This is confirmed by our experience. On November 30 and December 4, two articles referring to two other islands of this same archipelago were published in this same magazine, specifically in the Special Issue Genetic Resources for Viticulture. These documents contain very similar images.
For all these reasons, we ask you to accept this Figure as we present it.
Figure 8 = in panel a.2, please check the title “Nei Geneti 2K (6) ..”. It is not clear.
Modifications have been made. It is presented in the tab corresponding to this Figure of the "Manuscript Figures" Excel document.
Thank you so much
Table S1 = in the column “% Similarity to the nearest genome” a numeric value is expected, not a text; please correct.
The proposed change has been made.
Column “c” refers to colour of wine? If so, please add this word.
It does not refer to the color of the wine, but to the color of the vine variety. It is indicated in the same table. See for yourself:
What is the nearest genome? If the reference genome is not always the same, please include the info of reference genome for each entry.
The concept of "nearest genome" refers, as can be seen in the image, to the molecular profile of our database, which is either an identity, a coincidation in the 20 SSRs (100%), or it differs in the allelic length value of 1 allele (97.5% mutation), 2 (95% mutation), 3 (92.5% mutation), or 4 alleles (90% mutation). For the same variety it always refers to the same genetic profile. Reference molecular profiles are not shown because it is confidential information.
The field “different alleles” has only 4 columns: this means that new sizes were only reported for a maximum of four loci? Please explain it better.
This column only shows the differences, that is, the SSR alleles that present variation. We believe that it is not worth showing all 40 boxes since the boxes mostly coincide. We ask reviewer two to accept the Table as presented. It is a good formula to present information. This is demonstrated by our experience in publications. Since 2016 until December 4, we have published 6 articles on this topic in important Viticulture and similar journals. In all cases, the reviewers have praised the way of presenting the results in this Table.
Thank you so much.
In case your time allows, and you want to take a look at the articles published in our research group and for this topic, find the references below:
- Marsal, G.; Mateo, J.M.; Canals, J.M.; Zamora, F.; Fort, F. SSR analysis of 338 accessions planted in Penedes (Spain) reveals 28 unreported molecular profiles of Vitis vinifera Am. J. Enol. Vitic. 2016, 67, 466-470. https://doi.org/10.5344/ajev.2016.16013
- Marsal, G.; Bota. J.; Martorell, A.; Canals, J.M.; Zamora, F.; Fort, F. Local cultivars of Vitis vinifera in Spanish Islands: Balearic Archipelago. Sci. Hortic. 2017, 226, 122–132. https://doi.org/10.1016/j.scienta.2017.08.021
- Marsal, G.; Mendez, J.J.; Mateo-Sanz, J.M.; Ferrer, S.; Canals, J.M.; Zamora, F.; Fort, F. Molecular characterization of Vitis vinifera local cultivars from volcanic areas (the Canary Islands and Madeira) using SSR markers. Oeno One 2019, 4, 667-680. https://doi.org/10.20870/oeno-one.2019.53.4.2404
- Fort, F.; Marsal, G.; Mateo-Sanz, J.M.; Pena, V.; Canals, J.M.; Zamora, F. Molecular characterisation of the current cultivars of Vitis vinifera in Lanzarote (Canary Islands, Spain) reveals nine individuals which correspond to eight new varieties and two new sports. Oeno One 2022, 56, 281-295. https://doi.org/10.20870/oeno-one.2022.56.3.5519
- Fort, F.; Lin-Yang, Q.; Suárez-Abreu, L.R.; Sancho-Galán, P.; Canals, J.M.; Zamora, F. Study of Molecular Biodiversity and Population Structure of Vitis vinifera ssp. vinifera on the Volcanic Island of El Hierro (Canary Islands, Spain) by Using Microsatellite Markers. Horticulturae 2023, 9, 1297. https://doi.org/10.3390/horticulturae9121297
- Fort, F.; Lin-Yang, Q.; Valls, C.; Sancho-Galán, P.; Canals, J.M.; Zamora, F. Characterisation and Identification of Vines from Fuerteventura (Canary Volcanic Archipelago (Spain)) Using Simple Sequence Repeat Markers. Horticulturae 2023, 9, 1301. https://doi.org/10.3390/horticulturae9121301
Table S3 = The title should also report that the table includes size range (not rang) information and type of fluorescence label
You are right.
Annealing Temperature (Ta) of SSR.
We propose the following paragraph as an alternative to the existing one:
Annealing temperature (Ta), range (in base pairs) where the peaks have to appear in the electrophenogram and type of fluorescence label of the 20 SSRs used by TECNENOL.
Table S4 = in the title, please replace “characterization” with “characteristics”
You are right. This change has been made.
Comments on the Quality of English Language
I strongly recommend to check the English grammar (use of punctuation, parentheses) and pay particular attention to spot typos and Spanish “tracks”. Some sentences start with "and" and this should be avoided.
English grammar, punctuation and spelling has been double checked by authors. We hope to meet your expectations

Round 2
Reviewer 2 Report
Comments and Suggestions for Authors
Dear Authors
Thank you for your extensive replies to my comments to the manuscript. I see that some editing has been done to the manuscript; in other cases, you have provided your view on the points raised and decided to keep your initial choices. I am fine (in not always at all) with such decisions and decided not to go over with another round of revisions because the paper has all the elements required for publication. I believe that some of the replies you provided to me could have been of interest to other readers that could have asked the same questions after reading the paper; therefore, a clear explanation in the text of the choices made (in the M&M, Results or Discussion) could have been helpful. Similarly, the fact the “paper structure” has been already accepted several times is not a valid (despite a proof of evidence) element to keep going on in that way. Finally, I still think that some improvements could have been done to Figures 4b and 6b, but if the typesetting office is fine with that, is not a vital element.
Therefore, my decision is to accept the paper.